# Promotion Effect of Coexposure to a High-Fat Diet and Nano-Diethylnitrosamine on the Progression of Fatty Liver Malignant Transformation into Liver Cancer

**DOI:** 10.3390/ijms241814162

**Published:** 2023-09-15

**Authors:** Xin Yin, Yu-Sang Li, Sha-Zhou Ye, Ting Zhang, Yi-Wen Zhang, Yang Xi, He-Bin Tang

**Affiliations:** 1Lab of Hepatopharmacology and Ethnopharmacology, School of Pharmaceutical Sciences, South-Central Minzu University, No. 182, Minyuan Road, Wuhan 430074, China; kno521521@163.com (X.Y.); liys2006@mail.scuec.edu.cn (Y.-S.L.); kinima988@outlook.com (T.Z.); 2021120621@mail.scuec.edu.cn (Y.-W.Z.); 2Institute of Biochemistry and Molecular Biology, School of Medicine, Ningbo University, No. 818 Fenghua Road, Jiangbei District, Ningbo 315211, China; yeshazhou@foxmail.com

**Keywords:** high-fat diet, nano-diethylnitrosamine, fatty liver, liver cancer, transcriptomics

## Abstract

Overconsumption of high-fat foods increases the risk of fatty liver disease (FLD) and liver cancer with long pathogenic cycles. It is also known that the intake of the chemical poison nitrosamine and its nanopreparations can promote the development of liver injuries, such as FLD, and hepatic fibrosis, and significantly shorten the formation time of the liver cancer cycle. The present work confirmed that the coexposure of a high-fat diet (HFD) and nano-diethylnitrosamine (nano-DEN) altered the tumor microenvironment and studied the effect of this coexposure on the progression of fatty liver malignant transformation into liver cancer. Gene transcriptomics and immunostaining were used to evaluate the tumor promotion effect of the coexposure in mice. After coexposure treatment, tumor nodules were obviously increased, and inflammation levels were elevated. The liver transcriptomics analysis showed that the expression levels of inflammatory, fatty, and fibrosis-related factors in the coexposed group were increased in comparison with the nano-DEN- and high-fat-alone groups. The Kyoto Encyclopedia of Genes and Genomes (KEGG) results showed that coexposure aggravated the high expression of genes related to the carcinomatous pathway and accelerated the formation of the tumor microenvironment. The immunohistochemical staining results showed that the coexposure significantly increased the abnormal changes in proteins related to inflammation, proliferation, aging, and hypoxia in mouse liver tissues. The coexposure of high fat and nano-DEN aggravated the process of steatosis and carcinogenesis. In conclusion, the habitual consumption of pickled foods containing nitrosamines in a daily HFD significantly increases the risk of liver pathology lesions progressing from FLD to liver cancer.

## 1. Introduction

Liver cancer is one of the top cancer killers worldwide, with a high mortality rate. Globally, there are approximately 906,000 newly diagnosed cases of liver cancer, and it claimed the lives of over 830,000 people in 2020 [1]. The development of liver cancer is a complex process influenced by various risk factors. Among the main factors contributing to liver cancer are alcohol consumption, diabetes, obesity, and dietary exposures [2]. Substantial pieces of evidence also suggest that liver cancer develops through alterations (chromosomal alterations, genetic mutations, chromatin remodeling, histone modifications, DNA methylation, and expression levels) in certain Wnt signaling pathways [3]. These alterations affect genes that play a role in maintaining the normal structure and function of the liver but become modified during the process of liver carcinogenesis.

Furthermore, high-fat diets (HFD) and nitrosamines are both common factors in daily diets. In recent years, the consumption of high-fat ketogenic diets has gained increasing attention from many individuals, particularly those struggling with obesity, as a means to control cholesterol levels and achieve weight loss. However, there is much controversy surrounding the potential risks and effects of this approach. Dietary habits vary across countries, impacting the consumption and exposure to HFD and nitrosamines. For instance, China’s cuisine features fatty dishes such as stir-fries and deep-fried snacks, resulting in a high intake of fats and moderate exposure to nitrosamines from preserved foods. Similarly, South Korea’s traditional dishes such as samgyeopsal and bibimbap have a higher fat content, leading to moderate consumption of fats and nitrosamines. In contrast, the Inuit people in the Arctic follow an HFD primarily from marine mammals, but their remote location and traditional hunting practices limit their exposure to nitrosamines. It is intriguing to note that despite these protective factors, these countries still demonstrate an elevated susceptibility to fatty liver and cancer, as evidenced by the liver cancer incidence rates ranking provided by the International Agency for Research on Cancer (IARC/WHO). Therefore, conducting a comprehensive study on the simultaneous existence of HFD and nitrosamines is crucial for improving our understanding of cancer development mechanisms, devising effective prevention strategies, and promoting public health outcomes. Recent studies have reported that HFD and exposure to chemical toxic substances, which are key factors in the development of fatty liver disease (FLD), can induce obesity and insulin resistance. This results in the excessive lipids accumulation in the liver, creating an inflammatory microenvironment and ultimately promoting the progression from FLD to malignant liver cancer [4]. Long-term consumption of an HFD disrupts liver metabolism and promotes lipid accumulation, resulting in the phosphorylation of insulin signaling transcription pathways, insulin resistance, and altered expression of Akt, GLUT4, and lipogenic genes [5,6]. Insulin resistance triggers the breakdown of hepatic fat, with free fatty acids generated during this breakdown stimulating the expression of factors involved in cholesterol synthesis and fatty acid synthesis [7,8]. The accumulation of metabolized lipid substances and chemical toxins can cause lipid peroxidation, damage the cell membrane’s permeability, and lead to mitochondrial damage and inflammation, which further impairs the liver’s normal metabolism [9].

Hypertrophic adipocytes in the liver exhibit mitochondrial dysfunction and high levels of reactive oxygen species, leading to accelerated lipogenesis [10,11]. Excessive lipid accumulation in the liver leads to a hypoxic microenvironment within the tissue, activating hypoxia-inducible factor-1 (HIF-1) and promoting the expression of vascular endothelial growth factor to counteract hypoxia. Furthermore, it stimulates the aging of hepatic stellate cells and facilitates hepatic fibrosis [12]. In addition, liver steatosis and insulin resistance can also cause reactive oxygen species accumulation to produce oxidative stress and cause cell damage [13]. Cell injury activates various cyclin-dependent kinase inhibitors, enhances the secretion of proinflammatory factors, and induces gene changes that are resistant to apoptosis, leading to cell senescence [14]. As an important regulator of the cell cycle, p16 protein can stop the cell division process, leading to aging by binding to cyclin-dependent kinase inhibitors [14,15]. Notably, the expression level of p16 protein is low in normal liver tissue but upregulated in liver cancer tissue [16].

As a key factor of the Wnt signaling pathway, β-catenin plays an important role in the regulation of cell proliferation and apoptosis. Encoded by the *CTNNB1* gene, β-catenin informs part of the cadherin complex on the cell surface. Dysregulation of its activity is associated with hepatocellular carcinoma and other liver diseases. Notably, β-catenin participates in various signaling pathways, including TGF-β signaling, which promotes cell proliferation [17,18], and the FOXO signaling pathway in colorectal cancer cells [19]. When β-catenin is combined with HIF-1 under hypoxia, it induces the expression of genes that regulate adaptation and survival [20,21]. In addition, changes in the β-catenin signaling pathway may lead to the activation of hepatic stellate cells, which is necessary for fibrosis [22].

In previous work, we explored whether the chemical poison nitrosamine induced the accumulation of β-catenin in mice, and the nucleation rate was shown to be increased. Activating inflammatory factors include cyclooxygenase-2 (COX-2), tumor necrosis factor-α (TNF-α), and peroxisome proliferator-γ (PPAR-γ). Fibrosis-related factors include TGF-β1, which is involved in the malignant process of fatty liver [23,24]. These results indicate that β-catenin may activate multiple signaling pathways, resulting in the progression from FLD to liver cancer. In the previous study, we compared the mouse liver cancer model induced by nitrosamine and its nanopreparation and discovered that the nanopreparation rapidly induced liver cancer with significant efficacy and no side effects [25]. 

Long-term excessive intake of high-fat food is a well-known unhealthy eating habit associated with liver fatigue, metabolic disorders, and an elevated risk of developing fatty liver. Similarly, the prolonged consumption of nitrite-rich pickled food represents another detrimental habit that inflicts irreversible damage on the liver, thereby facilitating a more rapid and facilitated onset of liver cancer. Unfortunately, the extended duration required for lesion formation and the absence of symptoms affecting daily life make it easy to overlook the effects of pickled food consumption, inadvertently accelerating the occurrence of liver cancer without awareness.

In this study, we established a liver injury model of fatty liver malignant transformation into liver cancer by continuously feeding mice an HFD to simulate people’s daily HFD habits and explored the effect and mechanism of coexposure to high-fat and the chemical poison nano-diethylnitrosamine (nano-DEN) on liver cancer in mice. This can provide a reference for the early prevention and accurate treatment of FLD and its malignant transformation.

## 2. Results

### 2.1. Coexposure Phenotypically Aggravated Liver Cancer

This section aimed to investigate cancer progression through visual observation, employing the hematoxylin and eosin (H&E) staining method. Both macroscopic anatomical changes in the liver and microscopic pathological alterations in tissue sections were examined. Quantitative analysis was then conducted to evaluate the incidence and severity of liver tissue damage, which was scored based on three criteria: the number of individual tumor occurrences, inflammatory score of tumors, and hepatic steatosis (fat accumulation) score. As shown in Figure 1A, the liver surface of the normal group was smooth, soft, and bright, and white tumor nodules appeared on the mouse liver surface of the nano-DEN and high-fat groups. However, in the coexposure group, there were not only more white tumor nodules on the liver surface but also larger red blood clots. H&E staining (Figure 1B) showed that the liver tissue structure of control mice was normal, the lobules were complete, and the cell cord structure was arranged neatly. However, after stimulation and induction, the liver tissues of the other three model groups were damaged to different degrees. Fatty degeneration, loose cytoplasm, and irregular shapes appeared in the high-fat group. In the nano-DEN group, the nuclei were enlarged, some fat vacuoles appeared, and the inflammatory focus was clearly observed. However, the coexposure group showed the most serious phenotypic changes, with obvious fatty liver cells, the nucleus squeezed to one side by fat droplets, and a large number of inflammatory foci accompanied by tumor nodules. The results of tumor incidence at the 25th week showed that 41% of the total number of mice developed tumors from high-fat group (C+ group), 32% of the total number of mice developed tumors from nano-DEN group (M group). Importantly, the tumor incidence at the 25th week in the coexposure group (M+ group) was greater than 78%.

In our previous study, nano-DEN caused severe inflammatory infiltration during the development of liver cancer [25]. Therefore, to further understand the specifics of the risk factors for hepatic carcinogenesis, inflammation and steatosis scores are important indicators by which to judge the severity of liver tumors. In the present study, compared with the control group, the inflammation scores were increased in the high-fat group (2.13 ± 0.10 of control, *p* < 0.01) and the nano-DEN group (2.27 ±0.13 of control, *p* < 0.001). Surprisingly, the degree of inflammatory infiltration was also increased in the coexposure group (2.73 ± 0.13 of control, *p* < 0.001), and it was the most severe compared with the other two groups. Compared with the control group, the steatosis scores were increased in the high-fat group (2.40 ± 0.20 of control, *p* < 0.001) and the nano-DEN group (1.80 ± 0.37 of control, *p* < 0.01). Surprisingly, the degree of steatosis was also increased in the coexposure group (2.60 ± 0.24 of control, *p* < 0.001), and it was the most severe compared with the other two groups.

### 2.2. Effects of the Coexposure on Serum Biochemical Indices

The activities of glutamic oxaloacetic transaminase (GOT) and glutamic pyruvic transaminase (GPT) are important diagnostic indices of liver function. When the liver is substantially damaged, the concentration of both GOT and GPT will be upregulated. As shown in Table 1, we found that GOT (*p* < 0.01) and GPT (*p* < 0.05) were significantly increased in the nano-DEN group and reached the highest levels in the coexposure group. The ALP content of the nano-DEN group (*p* < 0.05, compared with the common group) and the coexposure group (*p* < 0.01, compared with the high-fat group) also increased abnormally. According to Table 1, we found that the TG content in the high-fat group and the coexposure group increased significantly (*p* < 0.01). Although there were no significant differences in serum total protein (TP) and total cholesterol (TCHO), both exhibited distinct trends after stimulation. The TP levels gradually decreased, whereas the TCHO levels showed a correlation with HFD consumption, leading to an increase. However, it is noteworthy that liver damage induced by nano-DEN actually resulted in a decrease in TCHO levels.

### 2.3. Transcriptomics Analysis of Liver Function in Mice after Coexposure

To enhance the assessment of gene expression related to these three dietary conditions at the molecular level and explore the impact of different diets on liver changes, we treated HFD, nano-DEN, and coexposure as distinct conditions. We aimed to identify significant variations in gene expression among these conditions and derive potential patterns of differences through comparison with the control group. The overall distribution of DEGs was inferred by volcano visualization. In the volcano plot of DEGs with *p*-value < 0.05 and |log2 fold change| > 0.585, the abscissa represents the log2 fold change, and the ordinate represents the −log10 (*p*-value). Each dot represents a specific gene or transcript. The red dots indicate significantly upregulated genes, blue dots indicate significantly downregulated genes, and black dots represent genes with no significant alteration. The most statistically significant genes (represented by dots) are toward the top right of the plot (Figure 2A). The top ten differential genes were screened by comparison with the control group. At 25 weeks, HFD, nano-DEN diet and coexposure diet all showed significant differences in gene expression compared with normal diet (Figure 2A), all with varying degrees of up- and down-regulation. Interestingly, a large proportion of liver cancer-related genes were more significantly altered in the coexposure group compared with the high-fat and nano-DEN groups (Appendix A), suggesting that coexposure may promote or suppress the expression of some key transcription factors, thereby accelerating liver cancer development.

The KEGG pathway analysis revealed a significant enrichment of DEGs in multiple metabolic pathways, including nonalcoholic fatty liver disease (NAFLD) and chemical carcinogenesis-reactive oxygen species (ROS) pathways, which are strongly associated with inflammation (Figure 2B). Particularly, it emphasizes that an HFD significantly enhances the activation of carcinogenesis-related signaling pathways, specifically the chemical carcinogenesis-ROS pathway and the NAFLD pathway, occupying prominent rank position of 4th and 8th, respectively. The network diagram results (Appendix A) further support these findings, indicating that the genes of these two pathways exhibited high expression levels in 43.5% (10/23) and 51.5% (17/33) of the samples, respectively. Interestingly, the p53 signaling pathway, FoxO signaling pathway, and AMPK signaling pathway, which are strongly linked to cancer suppression, demonstrated significant decreases in their rankings (to 16th, 38th, and 44th, respectively (Appendix A)). Furthermore, upon exposure to the chemical carcinogen nano-DEN, the pathways related to cancer suppression also became activated. Both the chemical carcinogenesis-ROS pathway and the NAFLD pathway were activated, ranking 11th and 15th, respectively (Appendix A). Additionally, tumor-suppressor genes were found to be upregulated, leading to the activation of associated pathways such as the FoxO signaling pathway, AMPK signaling pathway, and p53 signaling pathway. These pathways experienced substantial upregulation, occupying top positions (1st, 5th, and 6th, respectively) in the rankings. Moreover, when co-exposed to nano-DEN and an HFD, there was an increased activity in carcinogenesis-related signaling pathways, surpassing the effects of individual exposures. The chemical carcinogenesis-ROS pathway and the NAFLD pathway rose to the 2nd and 3rd positions, respectively, demonstrating greater activation compared to individual exposures. Notably, the genes associated with these two pathways exhibited remarkably high transcriptional expression levels of 62.1% (18/29) and 56.0% (14/25), respectively (Appendix A). Conversely, the signaling pathways associated with tumor-suppressor genes in the coexposure group were more suppressed compared to the nano-DEN group. The rankings of the p53 signaling pathway, AMPK signaling pathway, and FoxO signaling pathway underwent significant declines (to 19th, 49th, and 20th, respectively; Appendix A). These data suggest that increased expression of inflammation pathway-related genes in mouse liver creates a pro-inflammatory microenvironment. Simultaneously, there is a decrease in the expression of tumor suppressor genes in mice, increasing the likelihood of cancer development.

### 2.4. KEGG Pathway Enrichment Analysis of the Intersection of Genes in Mouse Liver Induced by Coexposure

Furthermore, we obtained liver tissues from mice in the HFD group, nano-DEN group, and coexposure group, in order to identify the enriched pathways that play a key role in the process of carcinogenesis. We drew a Venn diagram online (http://bioinfoogp.cnb.csic.es/tools/venny/index.html, accessed on 18 March 2022) and determined the common pathways of the 54 DEGs (Figure 3A). Subsequently, KEGG results showed that the differentially expressed genes were enriched in the phospholipase D signaling pathway, choline metabolism in cancer, and microRNAs in cancer (Figure 3B). The differentially expressed genes, including platelet-derived growth factor receptor α (PDGFRA: 219 ± 29% that of control, *p* < 0.01), tropomyosin 3 (TMP3: 233 ± 42% that of control, *p* < 0.05), and adenylate cyclase type 1 (ADCY1: 312 ± 68% that of control, *p* < 0.05), had the most significant increases in expression (Figure 3C). These results suggest that coexposure aggravates the high expression of genes related to the carcinomatous pathway and accelerates the formation of the tumor microenvironment.

### 2.5. KEGG Enrichment Analysis of Microenvironment-Associated Factors in Mouse Liver Tumors Induced by Coexposure

According to previous research in our laboratory [25], the development of FLD into liver cancer generally involves changes in tumor microenvironment-related pathways, including inflammation-related pathways, adipogenesis-related pathways, hypoxia-related pathways, and aging-related pathways. By analyzing the gene ratios (Table 2), we found that the differentially expressed genes in the coexposure group were more enriched in pathways related to inflammation (TNF signaling pathway), hypoxia (HIF-1 signaling pathway), and aging (cell cycle) than those in the nano-DEN group and high-fat group. These results suggest that the DEGs induced by coexposure are highly enriched in signaling pathways related to inflammation, hypoxia, and aging.

### 2.6. KEGG Functional Analyses of DEGs in the Liver Cancer Region of Mice Induced by Coexposure

Next, we collected liver cancer tissues from the mice in the nano-DEN group and the coexposure group. To focus on the changes occurring from precancerous to fully developed cancer stages, we compared the gene expression levels between the cancerous and paracancerous regions of liver tissues from mice of the coexposure group. The KEGG analysis revealed that DEGs induced by nano-DEN were enriched in several signaling pathways, including glycerophospholipid metabolism, ether lipid metabolism, glycosphingolipid biosynthesis lacto and neolacto series, fat digestion and absorption, and the cholesterol metabolism pathway (Figure 4B). These pathways play a crucial role in lipid metabolism reactions. Analysis of the gene ratios in Table 3 showed that the genes in the first four pathways mentioned above were significantly downregulated at the transcriptional level, while the genes in the ether lipid metabolism pathway were significantly upregulated (*p* < 0.05). Moreover, as shown in Figure 4C, the DEGs between the cancerous and paracancerous regions of liver tissues from mice of the coexposure group were mainly enriched in the NOD-like receptor signaling pathway, FoxO signaling pathway, and AMPK signaling pathway, which are closely associated with inflammatory responses. Additionally, based on Table 3, it was observed that 62.5% (5/8) of the differentially expressed genes related to the NOD-like receptor signaling pathway, which promotes inflammatory responses, showed high expression at the transcriptional level, while both the AMPK signaling pathway and the FoxO signaling pathway, which inhibit inflammatory responses, exhibited low expression in 80% (4/5) of the differentially expressed genes (*p* < 0.05). These results suggest that coexposure exacerbates the upregulation of inflammation-related genes in mouse liver tissue, promoting the malignant transformation of fatty liver into cancer. Furthermore, it was found that the differentially expressed genes between the cancerous and paracancerous regions of liver tissues from mice of the coexposure group were enriched in inflammation (IL-17 signaling pathway, TGF-β signaling pathway, PPAR signaling pathway, and TNF signaling pathway), hypoxia (HIF-1 signaling pathway), and aging (cell cycle)-related pathways. The above results suggest that coexposure aggravated the high expression of genes related to inflammation, hypoxia, and aging pathways in the mouse liver, promoting the malignant transformation of fatty liver into cancer.

### 2.7. Coexposure Significantly Increased the Protein Expression of COX-2, β-Catenin, HIF-1α, and p16 in the Mouse Tumor Microenvironment

To further demonstrate the effectiveness of coexposure, we monitored the changes in several pivotal factors in the tumor microenvironment [26], including COX-2, β-catenin, HIF-1α, and p16 via protein immunostaining of the livers of mice in each group.

From the immunostaining results, we found that coexposure induced inflammation in the liver tissues of mice. The results (Figure 5) showed that COX-2 protein expression in the livers of mice of the coexposure group increased significantly (278% ± 16% that of control, *p* < 0.05). Abnormal accumulation of β-catenin is considered to be very important in the development of liver cancer. In Figure 5, in the liver of the coexposure induced mice, the β-catenin expression was abnormally increased in the cytoplasm (242% ± 18% that of control, *p* < 0.001) and nucleus (234% ± 18% that of control, *p* < 0.001) of hepatocytes. HIF-1α is a key transcriptional regulator in the cell oxygen signaling pathway that can activate downstream target genes to cause the proliferation, metastasis, and invasion of tumor cells and accelerate disease progression. As depicted in Figure 5, the protein expression level of HIF-1α in the livers of the coexposure group showed a significant increase (242% ± 18% of the control, *p* < 0.001) compared to the control group. Additionally, p16 protein plays a role in promoting cell proliferation and malignant transformation, and its inactivation leads to cellular aging. In Figure 5, it can be observed that the protein expression of p16 in mouse liver tissue significantly increased in the coexposure group (240% ± 23% of the control, *p* < 0.05) when compared to the control group.

## 3. Discussion

In recent years, increasing evidence has shown that liver injury is the result of metabolic stress or exposure to toxic substances and microbial pathogens, which is characterized by liver cell injury and death and inflammatory cell infiltration [27]. However, disorder of this process often promotes the formation and development of liver cancer [28]. Countless external stimuli can damage the liver in daily life, all of which can promote the occurrence of liver cancer [29]. Taking the recent popular ketogenic diet as an example, it is a high-fat dietary habit. This diet may have some benefits for certain populations such as individuals seeking continued weight loss, type 2 diabetes patients, and those with neurodegenerative diseases (such as epilepsy and Parkinson’s disease). However, it may not be suitable for individuals with liver disease, pancreatitis, or gallbladder disease. Furthermore, the ketogenic diet imposes strict requirements on carbohydrate intake, which must be maintained within a very limited range. Failure to properly control carbohydrate intake may increase the burden on the liver and worsen liver damage. Additionally, if individuals following an HFD, including those on the ketogenic diet, also have a habit of consuming foods with high levels of nitrites, such as Korean kimchi, salted fish, bacon, and other preserved products, this undoubtedly accelerates and exacerbates the formation of FLD and its progression to cancer. We believe that every dietary approach has pros and cons, and the key lies in balancing and choosing wisely. It is essential to consult with professionals and select a diet that suits one’s individual needs. In addition, considerable study has suggested that sex difference plays an important role in the progression of liver damages, ranging from virus-induced liver injuries, hepatitis, and hepatocirrhosis, to liver cancer. Therefore, in this experiment, male mice were given an HFD and intraperitoneal injection of nano-DEN to induce a liver cancer model.

To visualize and compare the differences in mice, we generated a schematic diagram that ranks the activation status of selected signaling pathways in our experiments, incorporating both previous findings and our current results (Figure 6). Previous experiments observed that continuous feeding of mice with an HFD resulted in significant liver damage, including steatosis, in 41% of the mice at 25 weeks, fibrosis and cirrhosis in 61% of the mice at 35 weeks, and hepatocellular carcinoma in 89% of the mice at 52 weeks [30]. Consistent with our previous studies, when mice were solely exposed to nano-DEN, 61% of the mice exhibited evident pathological signs of cancer at 25 weeks, and 62.5% of the mice developed liver cancer at 35 weeks [23]. To enable comparison, we replicated their previous dosing and divided the mice into three groups, administering different treatments while conducting RNA sequencing at 25 weeks. Notably, when mice were coexposed to an HFD and nano-DEN, the time for the development of liver cancer was significantly shortened to 25 weeks compared to the other two groups. Through gene sequencing of liver tissues from three groups of mice, we made an interesting discovery. The consumption of an HFD significantly enhanced the activation of oncogenic signaling pathways, with particular emphasis on the NAFLD and chemical carcinogenesis-ROS, which ranked prominently. Intriguingly, the ranking of cancer suppressor-related pathways such as the p53, FoxO, and AMPK pathways exhibited a significant decrease in comparison. Moreover, upon exposure to nano-DEN, we observed the activation of pathways associated with cancer inhibition. Tumor suppressor genes were found to be upregulated, consequently activating pertinent pathways, including FoxO, AMPK, and p53. Remarkably, when mice were coexposed to both nano-DEN and an HFD, the activity of carcinogenesis-related signaling pathways increased significantly, surpassing the effects seen in the individual treating groups. In comparison to the nano-DEN group, the coexposure group exhibited a greater suppression in cancer suppressor-related pathways. These findings shed light on the potential implications. It appears that an HFD, particularly when combined with the exposure to nano-DEN, may compromise the body’s natural self-repair mechanisms and its ability to inhibit cancer development. This phenomenon can be attributed to the suppression of cancer suppressor-related pathways, resulting in decreased activity of genes involved in these pathways. Consequently, the body’s response to carcinogens is accelerated, potentially exacerbating the risk of cancer. In the present work, we focused on critical proteins related to cancer development and progression and identified three representative protein pathways associated with the signaling pathways. Through subsequent experimental analysis, we examined the expression patterns, interactions, and functional roles of these key proteins. Our results enhance the understanding of the complex molecular mechanisms underlying the observed phenomena and strengthen the validity of our conclusions. By utilizing a range of techniques such as HE staining, immunohistochemical staining, and quantitative analysis using the Nuance multispectral imaging system, we were able to precisely identify and quantitatively analyze the alterations occurring within these pathways. This comprehensive approach bolstered the support for our conclusions.

Based on the H&E staining chart, we evaluated and analyzed the degree of liver injury in experimental mice. The results showed that both the high-fat group and the nano-DEN group had different degrees of structural disorder and irregular shapes of liver tissue, accompanied by a few tumor nodules and inflammatory foci. The liver tissue of mice in the exposed group showed a large number of fatty vacuoles, inflammatory foci, and tumor nodules (Figure 1A). 

Alkaline phosphatase (ALP) mainly passes through the hepatobiliary system and is then excreted through bile. If the liver is damaged, causing inflammation and fatness, metabolism is abnormal, which leads to an abnormal increase in ALP concentration in the blood. Triglycerides are important molecules used by the human body to store energy, too. Once the liver is fatty and inflamed to different degrees, the content of serum triglycerides obviously increases. When the liver is damaged, the level of these three factors in the serum increases on average. Therefore, serum levels of GOT, GPT, and ALP are important diagnostic indicators of liver function. The serum data of this experiment showed that compared with the high-fat group and nano-DEN group, the levels of these three indices in the serum of the coexposure group were significantly higher. The results suggest that the coexposure of high-fat and nano-DEN aggravates the damage to liver tissue structure, promotes the formation of fatty liver disease, and accelerates its malignant transformation into liver cancer. It has been reported that overconsumption of an HFD causes an imbalance between lipid intake and output and in the deposition and removal of fat [31], resulting in excessive accumulation of lipids in the liver. An abnormal increase in serum TG levels is the manifestation of lipid imbalance [8]. In this experiment, continuous high-fat feeding caused a significant increase in serum TG levels in mice, and it was higher in the coexposure group. From the above data (Table 1), we know that coexposure promotes disordered lipid metabolism in the liver, excessive accumulation of fat, steatohepatitis, and the deterioration of the microenvironment in the liver.

In terms of mechanism, sustained hepatocyte cell death leads to the activation of immune cells in the liver, which gradually produces inflammatory cytokines and an inflammatory microenvironment related to the formation of liver cancer [32,33,34,35,36,37,38]. Consistent with the literature report, the inflammatory score of the pathological staining patterns of mice in the coexposure group in this experiment was indeed the highest (Figure 1B). We also identified the top ten differentially expressed genes when compared to the control group. At the end of the experiment, we observed significant differences in gene expression among the HFD, nano-DEN diet, and coexposure diet groups when compared to the normal diet (Figure 2). The coexposure group exhibited a larger number of liver cancer-related genes with more significant changes compared to the high-fat group and nano-DEN group (Figure 3, Appendix A). This suggests that coexposure may have the potential to promote or inhibit the expression of key transcription factors, thereby accelerating the development of liver cancer. Compared with the high-fat group and nano-DEN group, the differentially expressed genes in the liver tissue of the coexposure group were enriched in the TNF signaling pathway (inflammatory pathway) to a higher degree. Compared with the nano-DEN group, the differential genes of the cancer region in the liver tissue of coexposed mice were more enriched in the IL-17 signaling pathway, TGF-β signaling pathway, and PPAR signaling pathway (Figure 4). The above results suggest that coexposure exacerbated the high expression of genes associated with inflammatory pathways in mice livers and promoted the malignant transformation of FLD into cancer. In our comprehensive data and visual analysis, we have identified distinct alterations in inflammation-related pathways, hypoxia-related pathways, and aging-related pathways that exhibit pronounced significance in the progression from liver injury to hepatocellular carcinoma. Considering the pivotal protein components associated with inflammation factors (COX-2 and β-catenin), hypoxia (HIF-1α), and aging-related factors (p16), we regarded these as indicators for evaluating the extent of liver carcinogenesis. Therefore, we performed IHC staining for COX-2, β-catenin, HIF-1α, and p16 proteins to further substantiate our inferences from pathological aspects (Figure 5).

Our previous study also identified an inflammatory loop between β-catenin signal activation and COX-2, which is involved in the formation and maintenance of the inflammatory microenvironment of cells [39]. Similarly, the expression levels of COX-2 and β-catenin in the liver tissue of coexposed mice were the highest compared with the high-fat group and nano-DEN group (COX-2: 200 ± 19% of nano-DEN; 151 ± 23% of HFD; β-catenin: 179 ± 22% of nano-DEN; 140 ± 22% of HFD). All the above findings indicate that coexposure can accelerate the activation of the expression of inflammation-related pathway factors, promote the formation of the tumor microenvironment, and lead to liver cancer. Studies have shown that the production of ROS in the injured liver can lead to DNA damage and epigenetic modification, which activate oncogenes or inactivate tumor suppressor factors [40]. In our experiment, we also confirmed that compared with the high-fat group, the high expression of differentially expressed genes enriched in the chemical carcinogen-reactive oxygen species pathway in the coexposure group was more significant. At the same time, lipid peroxidation in the liver causes hypoxia [41,42], which leads to the overexpression of the hypoxia factor HIF-1 and further leads to an imbalance of the internal environment in the liver. At the same time, the sequencing results also showed that the differentially expressed genes in the coexposure group were distributed in the HIF signaling pathway, and the enrichment degree was higher than that in the nano-DEN group. Compared with the nano-DEN group and HFD group, HIF-1α was expressed at higher levels in the coexposure group (169 ± 20% of nano-DEN; 126 ± 23% of HFD), which was consistent with the literature report. The above findings further confirm that coexposure can aggravate tissue hypoxia and cell cancerization.

Once the tissue is deprived of oxygen, the aging of cells is accelerated, causing the transcription of factors related to the cell cycle to change. As a cell cycle regulatory protein, p16 can prevent CDK4/6 phosphorylation of regulatory cellular proteins [43]. In this part of the experiment, compared with the nano-DEN group, the differentially expressed genes in liver tissues of coexposed mice were more enriched in the cell cycle pathway. Compared with the nano-DEN group and HFD group, the protein staining results also showed that p16 protein was more highly expressed in the liver tissue of coexposed mice (193 ± 30% of nano-DEN; 163 ± 31% of HFD). Therefore, we can hypothesize that coexposure can accelerate the change in the cell cycle, promote cell aging, cause malignant proliferation, and lead to cancer.

Based on the above findings, we found that coexposure exhibited worse cancer effects in regard to the expression of some proteins and mRNAs. Although we have found evidence that coexposure can aggravate the formation of the tumor microenvironment, whether it is the only method of liver cancer formation requires further study.

## 4. Material and Methods

### 4.1. Materials

The preparation of the nano-DEN solution containing 41.25 mg/mL diethylnitrosamine [12] used in this experiment was consistent with our previous report [23]. DEN was provided by Tokyo Chemical Industry Co., Ltd. (Tokyo, Japan). Tween-80 was purchased from Amresco^®^ (Amresco, Solon, OH, USA). Lecithin from egg yolk and sesame oil were obtained from Sinopharm Chemical Reagent Co., Ltd. (Shanghai, China). Ultrapure deionized water was used for all experiments. In addition, the control group and nano-DEN group were provided with regular feed, while the high-fat group and coexposure group were given obesity model series diets (DIO series Diets). The regular feed had a fat energy supply rate of 10%, whereas the HFD had a fat energy supply rate of 60%. Both the regular feed and high-fat feed were sourced from Beijing Huafukang Biotechnology Co., Ltd. (Beijing, China). The composition of the HFD for mice (No. H10060) consisted of casein, dextrin, sucrose, cellulose, soybean oil, lard, various minerals, various vitamins, and choline, with fat accounting for 60% of the total energy intake, carbohydrates accounting for 20%, and protein accounting for 20%. The antibodies (Abcam Inc., Cambridge, UK) used in this study included antibodies against β-catenin (dilution of 1:200), HIF-1α (dilution of 1:200), p16 (dilution of 1:200), and COX-2 (dilution of 1:200).

### 4.2. Animal Care

Before performing the experiments, 20 male Kunming mice (18–22 g) from the Hubei Experimental Animal Center were acclimatized for 7 days under Specific Pathogen Free (SPF) conditions. All research on mice was approved by the Animal Experiment Ethics Committee of the South-Central Minzu University, Wuhan, China (permit number: 2019-SCUEC-AEC-013).

### 4.3. Animal Model Establishment 

In brief, 20 male Kunming mice were randomly divided into four groups (*n* = 5/group): the control group, nano-DEN group, high-fat group, and coexposure group. Among them, the common group and nano-DEN group were fed common feed and the high-fat group and coexposure group were fed a high-fat diet. During the experiment, the animals in the nano-DEN and coexposure groups were injected intraperitoneally with the chemical poison nano-DEN (16.5 mg/kg) weekly. Animals in the normal group and the HFD group were given an intraperitoneal injection of the same amount of normal saline weekly. At the 25th week of the experiment, all mice were sacrificed, and their liver tissues were collected.

### 4.4. Histopathological Analysis of Liver Tissues

Fresh liver tissues were washed with normal saline, fixed in formalin for 24 h, and placed in an automatic dehydrator for dehydration for 18 h. These tissues were then embedded in liquid paraffin to cool, cut into 2 μm sections, and stained with H&E. Tumor incidence represents the percentage of mice in each group with visible surface-hemorrhaging liver tumors (≥0.5 mm in diameter) [44]. The inflammation score statistics were obtained from the H&E-stained sections [45]. The inflammation score standard was defined as follows with a 40× objective lens: 0 lesions = 0 points; 1 lesion = 1 point; 2 lesions = 2 points; and 3 or more lesions = 3 points. For hepatocellular steatosis, specimens were scored 0–3 points (0 points: no fat; 1 point: steatosis occupying <33% of the hepatic parenchyma; 2 points: 34–66% of the hepatic parenchyma; and 3 points: more than 66% of the hepatic parenchyma [46]).

### 4.5. Sample Collection and Quality Control

Simultaneously with the pathological examination group, another portion of each liver tissue sample was processed for whole-transcriptome analyses. Immediately after dissection, the liver tissue was cut into small chunks, approximately the size of a soybean. Each chunk was rapidly placed into pre-labeled vials of liquid nitrogen to ensure immediate freezing for subsequent whole-transcriptome analyses. Total RNA was extracted from liver and tumor tissues collected from a total of 20 mice. The quality of RNA samples was evaluated using the Agilent Bioanalyzer to ensure they met the experimental requirements.

### 4.6. RNA Library Preparation and Sequencing Platform and Method

RNA libraries were prepared following the standard protocols provided by Illumina. This involved converting the RNA samples into libraries suitable for sequencing. Libraries were sequenced on Illumina HiSeq 4000 platforms (Origingene, Beijing, China). The reference genome used was sourced from the Ensembl database, with the genome version being GRCm38, and the gene annotation information was based on Ensemble 92. The high-quality sequencing reads obtained after quality control were processed using STAR software (version 2.7.7a) to align the read sequences with the specified reference genome (GRCm38) for the study species, which was the mouse (*Mus musculus*).

### 4.7. Identification of Differentially Expressed Genes (DEGs) 

DESeq2 was used for analyzing differential gene expressions, enabling the identification of genes that exhibited statistically significant differences between the two sample groups. DEGs were identified using DESeq2 (version 1.41.2, *p* value < 0.05 and |log2 fold change| > 0.585) [47].

### 4.8. KEGG Pathway Enrichment Analysis

The R package “clusterprofiler” was used to conduct KEGG pathway enrichment analysis, which helped identify the enrichment of DEGs in metabolic pathways and biological functions [48]. By utilizing the KEGG database, the genes from the samples were categorized based on their involvement in specific pathways or functional categories. The differential genes between pairwise groups, with one sample serving as the control, were then visualized on KEGG pathway maps, showcasing the annotated pathways specific to these differential genes. The calculation was performed using Fisher’s exact test. To control the false discovery rate (FDR) during the calculations, the Benjamini–Hochberg (BH) method was employed for multiple testing. The corrected *p*-values were set at 0.05, and KEGG pathways that met this criterion were defined as significantly enriched in differentially expressed genes.

### 4.9. Data Deposition and Accessibility

The data generated from this study has been deposited into the CNGB Sequence Archive, the database of the China National GeneBank [49]. It has been assigned the accession number CNP0003645, making it accessible to other researchers for validation and replication of the research findings [50].

### 4.10. Immunohistochemical Analysis of the Liver Tissues

The paraffin sections were deparaffinized with xylene, dehydrated with gradient alcohol, and incubated with 3% hydrogen peroxide for approximately 15 min. Then, the samples were repaired in citrate buffer that was diluted with ddH_2_O in a microwave oven for 10 min to perform antigen retrieval. After blocking with 5% bovine serum albuminin in a 37 °C oven for 1 h, the slices were combined with anti-β-catenin (dilution of 1:200), anti-COX-2 (dilution of 1:200), anti-HIF-1α (dilution of 1:200) and anti-p16 (dilution of 1:200) primary antibodies overnight. After the corresponding secondary antibody was added directly to the sample, it was incubated at 37 °C for 1 h. Then, protein expression was observed under a microscope after staining with diaminobenzidine chromophore. The nuclei in the tissue sections were counterstained with hematoxylin. The samples were washed with ddH_2_O to remove the dye solution, returned to a blue color in warm water for 5–10 min, and finally immersed in gradient ethanol for dehydration and mounting. As described in our previous study, the Nuance multispectral imaging system (Cambridge Research and Instruments, Woburn, MA, USA) was used to obtain multispectral images and perform quantitative analysis [51].

### 4.11. Statistical Analysis

The experimental data are presented as the means ± SEM, and each group in the experiment consists of five samples (*n* = 5). There are a total of four groups. The GraphPad Prism 9.0 (GraphPad Software, Inc., San Diego, CA, USA) scientific statistical software was used to analyze the experimental data and graphs. All statistical investigations of the differences among the groups were analyzed via one-way ANOVA, which was followed by Bonferroni’s post hoc tests. The statistical graphs are drawn based on the percentage of the expression of each group of proteins compared to the average value of the expression of the control group. Statistically, if *p* was less than 0.05, the difference was considered significant. The levels of differences are marked in each legend in the chart, as shown below: compared with the control group, *, *p* < 0.05, **, *p* < 0.01, ***, *p* < 0.001; compared with the nano-DEN group, #, *p* < 0.05, ##, *p* < 0.01, ###, *p* < 0.001. 

## 5. Conclusions

In this study, we obtained molecular findings that sustained simultaneous consumption of pickled food containing nitrosamine. A daily high-fat diet, on the one hand, injures the hepatocytes, resulting in the release of inflammatory factors to change the microenvironment of cells; on the other hand, cellular signaling altered by the coexposure promotes cell aging and epigenetic modification to accelerate tumor development. Thus, our research model provides important basic clues for liver injury and tumorigenesis, which is valuable for the early prevention or therapy of liver disease.

## Figures and Tables

**Figure 1 ijms-24-14162-f001:**
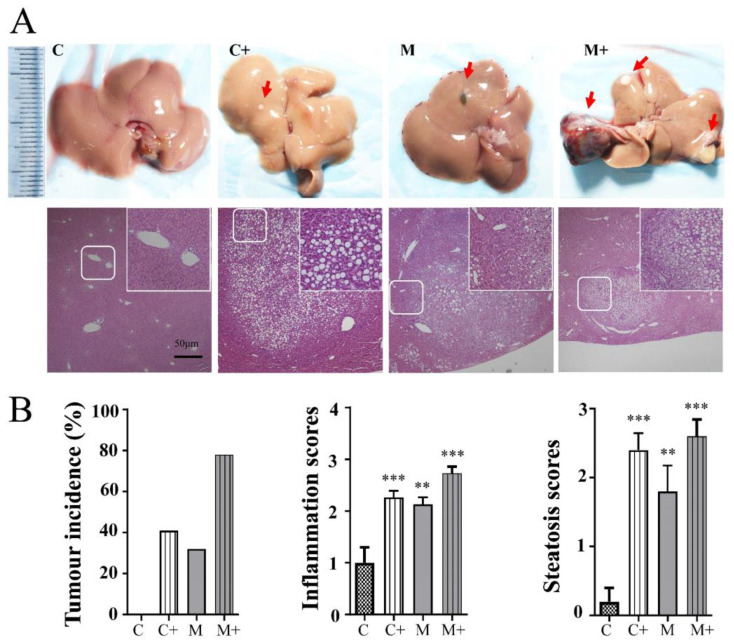
Coexposure aggravated the effect of fatty liver malignant transformation into liver cancer in mice. (**A**) A general view of the liver (with a standard steel ruler in centimeters) of the control group (C), high-fat group (C+), nano-DEN group (M), and coexposure group (M+) and the corresponding HE-stained pictures (10× objective lens). The upper right corner is a partially enlarged view of the picture (40× objective lens). The scale bar is 50 μm for all images. Red arrows point to the tissue in the liver where macroscopically visible liver tumor nodules have formed. (**B**) The incidence of tumors in each group of mice. The inflammation scores of the liver samples in the H&E-stained images. The red arrow points to the tumor nodule. The results are expressed as the means ± SEM. ** and *** denote *p* < 0.01 and *p* < 0.001 in comparison to the control.

**Figure 2 ijms-24-14162-f002:**
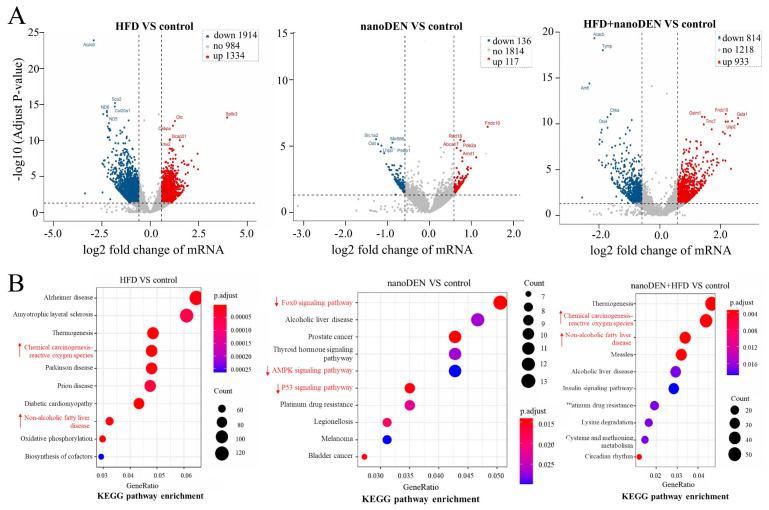
Visual analysis of DEGs and KEGG pathway enrichment analysis. (**A**) Volcano plot of DEGs of the HFD group (HFD), nano-DEN group (nano-DEN), and coexposure group (HFD + nano-DEN) in liver tissues. Red dots represent significantly upregulated genes, blue dots represent significantly downregulated genes, and black dots represent nonsignificant genes. (**B**) KEGG pathway enrichment analysis of DEGs in the liver tissues of mice in the HFD group, nano-DEN group, and coexposure group. The *x*-axis represents the enrichment factor, and the *y*-axis represents the pathway terms. The diameters of the circles represent the number of DEGs in the pathway. The color gradient on the right represents the *p*-value. The symbol ↑ highlighted in red indicates an increase in ranking position, ↓ indicates an decrease.

**Figure 3 ijms-24-14162-f003:**
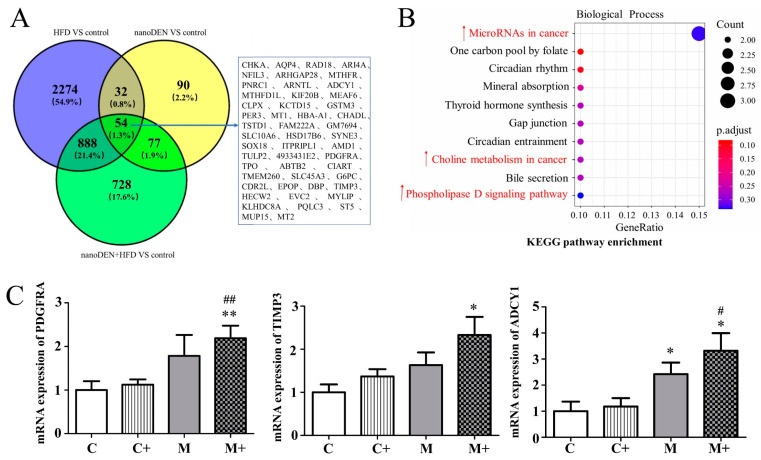
KEGG pathway enrichment analysis of common genes and mRNA expression levels of carcinomatous-related factors. (**A**) Common genes of the three DEGs in Figure 2 shown as a Venn diagram. (**B**) KEGG pathway enrichment analysis of the common genes. The symbol ↑ highlighted in red indicates an increase in ranking position. (**C**) Relative mRNA expression of *PDGFRA*, *TIMP3*, and *ADCY1* in liver tissue. Data are presented as the means ± SEM. * and ** denote *p* < 0.05 and *p* < 0.01 in comparison to the control group (C); # and ## denote *p* < 0.05 and *p* < 0.01 in comparison to the nano-DEN group (M).

**Figure 4 ijms-24-14162-f004:**
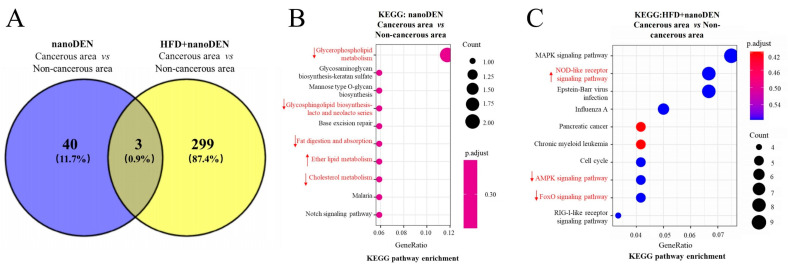
KEGG pathway enrichment analysis of DEGs in the cancerous region. The symbol ↑ highlighted in red indicates an increase in ranking position, ↓ indicates an decrease. (**A**) Common genes of the two DEGs in the cancerous region are shown as a Venn diagram; KEGG pathway enrichment analysis of the (**B**) nano-DEN group and (**C**) coexposure group in the cancerous region of liver tissues.

**Figure 5 ijms-24-14162-f005:**
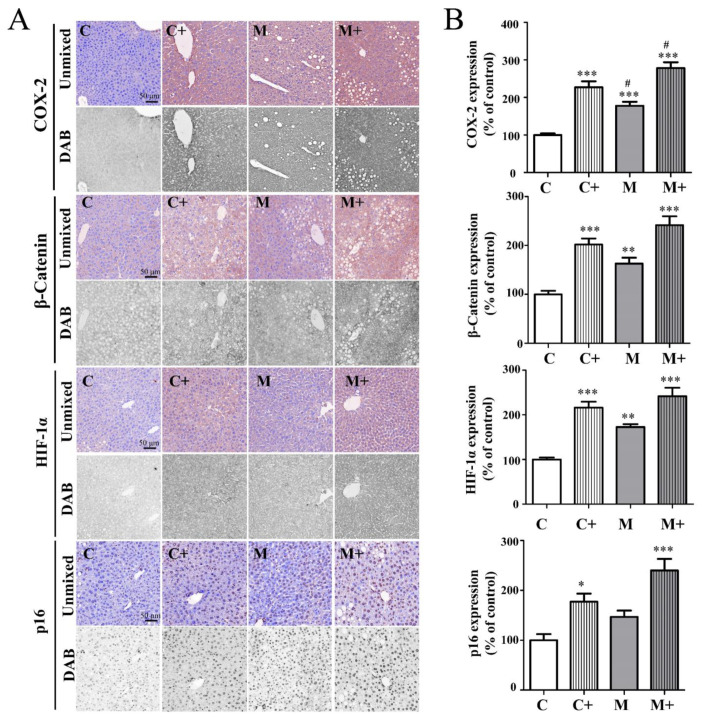
Coexposure aggravated the overexpression of COX-2, β-catenin, HIF-1α, and p16 proteins in the liver tissue. (**A**) Expressions of COX-2, HIF-1α, and p16 determined by immunohistochemistry. The scale bar is 50 μm for all images (20× objective lens). (**B**) Quantitative expression of COX-2, HIF-1α, and p16 determined by immunohistochemistry. The results are expressed as the means ± SEM. *, ** and *** denote *p* < 0.05, *p* < 0.01 and *p* < 0.001 in comparison to the control group (C); # denotes *p* < 0.05 in comparison to the nano-DEN group (M).

**Figure 6 ijms-24-14162-f006:**
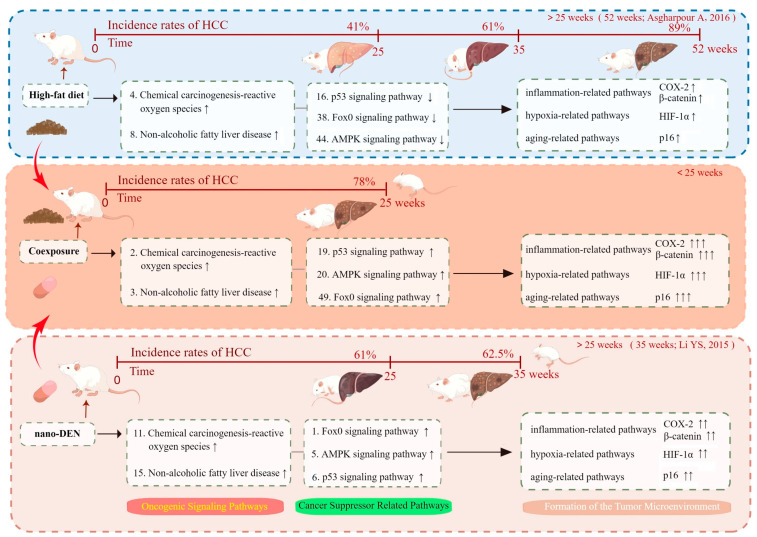
Mechanisms of carcinogenesis based on a comprehensive review of the literature from Refs. [23,30]. The symbol ↑ indicates an increase in ranking position, ↓ indicates an decrease.

**Table 1 ijms-24-14162-t001:** Serum biochemical indicators of mice after 25 weeks of exposure.

	GOT (U/L)	GPT (U/L)	ALP (U/L)	TG (mg/dL)	TP (mg/dL)	TCHO (mg/dL)
C	109.46 ± 28.34	48.58 ± 18.41	71.50 ± 32.75	132.38 ± 47.90	5.85 ± 0.40	115.23 ± 34.19
M	118.00 ± 37.90 *	69.85 ± 20.58	61.38 ± 40.72	184.15 ± 37.82 **	5.74 ± 0.26	126.13 ± 18.87
C+	220.73 ± 20.90 ^##^	123.45 ± 31.75 ^#^	107.27 ± 38.41 ^#^	136.91 ± 45.12	5.43 ± 0.51	118.09 ± 32.22
M+	237.36 ± 20.90 ^##^	142.73 ± 32.50 ^#^	118.55 ± 33.08 ^##^	187.36 ± 29.36 **	5.25 ± 0.26	126.73 ± 19.81

Note: Data are presented as the mean ± SEM. * and ** denote *p* < 0.05 and *p* < 0.01 in comparison to the control group (C); # and ## denote *p* < 0.05 and *p* < 0.01 in comparison to the HFD group (C+). Abbreviations: GOT, glutamic oxaloacetic transaminase; GPT, glutamic pyruvic transaminase; ALP, Alkaline phosphatase; TG, Triglyceride; TP, total protein; TCHO, total cholesterol.

**Table 2 ijms-24-14162-t002:** KEGG enrichment analysis of different target genes in each group.

Term	ID	HFD vs. Control				Nano-DEN vs. Control				HFD + Nano-DEN vs. Control			
		Gene Ratio	Related Targets	*p* Value	Rank	Gene Ratio	Related Targets	*p* Value	Rank	Gene Ratio	Related Targets	*p* Value	Rank
Chemical carcinogenesis-reactive oxygen species	mmu05208	92/1917	Ndufab1/Ndufs3/ Ndufb10/Ndufc2/ Ndufb4/Atp5 g……	8.03 × 10^−12^	4	12/257	Ndufa12/As3mt/Map2k2/ Ndufs1/Mgst2/Pik3ca/ Ikbkb/Hgf/Gstt2/ Gstm3/Gsta1/Cox5b	2.64 × 10^−2^	11	48/1094	Ndufa12/As3mt/Map2k2/ Ndufs1/Mgst2/Pik3ca// Gsta1/Cox5b……	4.74 × 10^−5^	2 ↑
Wnt signaling pathway	mmu04310	45/1917	Daam2/Senp2/Chd8/ Prickle3/Znrf3……	6.14 × 10^−2^	47	2/257	Fzd7/Ccnd1	9.58 × 10^−1^	248	19/1094	Daam2/Senp2/Cby1/ Nkd2/Rbx1/Cxxc4……	6.89 × 10^−1^	89
Cell cycle	mmu04110	34/1917	Anapc15/Cdc16/ Anapc5/Mad2l1……	7.34 × 10^−2^	89	6/257	Gadd45 g/Wee1/Tfdp2/ Orc2/Mdm2/Ccnd1	1.51 × 10^−1^	98	19/1094	Rbx1/Anapc4/Hdac1/ Cul1/Smc1a/Cdc14a……	1.87 × 10^−1^	88 ↑
HIF-1 signaling pathway	mmu04066	33/1917	Rps6 kb1/Vhl/Trf/ Rps6 kb2/Slc2a1……	3.53 × 10^−2^	93	4/257	Map2k2/Tek/Pik3ca/Eif4e	4.16 × 10^−1^	166	20/1094	Mtor/Rbx1/Tlr4/ Stat3/Rps6/Serpine1……	6.02 × 10^−2^	82
TNF signaling pathway	mmu04668	25/1917	Tab 2/Tab 1/Ripk3/ Mapk12/Mapk13……	4.66 × 10^−1^	135	6/257	Pik3ca/Mmp9/Lif/ Ikbkb/Creb3/Casp3	1.07 × 10^−1^	95	20/1094	Tab 2/Tab 3/Map3k5/ Map2k6/Vcam1……	5.58 × 10^−2^	83 ↑
PPAR signaling pathway	mmu03320	21/1917	Cpt1c/Pck2/Angptl4/ Fads2/Acsl5……	3.49 × 10^−1^	163	3/257	Plin5/Angptl4/Cpt1b	4.74 × 10^−1^	190	7/1094	Plin4/Sorbs1/Me1/ Fabp5/Fabp1/Cpt1b/Acox1	9.31 × 10^−1^	227
Fatty acid metabolism	mmu01212	21/1917	Echs1/Cpt1c/Fads1 /Hacd2/Fads2……	1.59 × 10^−2^	160	2/257	Cpt1b/Elovl3	5.36 × 10^−1^	231	6/1094	Fads1/Hacd2/Elovl6/ Cpt1b/Acox1/Acat2	7.87 × 10^−1^	244
TGF-β signaling pathway	mmu04350	19/1917	Ppp2r1b/Rps6 kb1/ Hfe2/Rps6 kb2……	6.74 × 10^−1^	185	3/257	Ppp2r1b/Smad7/Dcn	5.17 × 10^−1^	194	8/1094	Ppp2r1b/Rgmb/Rbx1/ Cul1/Neo1/Smad7/Id1/Chrd	9.09 × 10^−1^	213
IL-17 signaling pathway	mmu04657	15/1917	Tab 2/Anapc5/ Il17rb/Mapk15……	9.04 × 10^−1^	225	7/257	Mapk15/Mmp9/Il17rc/ Lcn2/Ikbkb/Fosb/Casp3	1.56 × 10^−2^	74	12/1094	Tab 2/Tab 3/Il17rb/ Traf2/Tnfaip3/Nfkbia/ Nfkb1/Il1b/Fosb/ Fos/Cebpb/Traf3ip2	4.38 × 10^−1^	167
VEGF signaling pathway	mmu04370	15/1917	Mapk12/Nos3/Kdr/ Rac1/Pik3cd……	2.48 × 10^−1^	222	5/257	Map2k2/Pik3ca/Mapkapk2/ Casp9/Mapkapk3	2.51 × 10^−2^	118	7/1094	Pla2g4f/Pik3ca/Nos3/ Mapkapk2/Ptk2/ Akt2/Raf1	5.76 × 10^−1^	227

The symbol ↑ highlighted in red indicates an increase in ranking position.

**Table 3 ijms-24-14162-t003:** KEGG enrichment analysis of different target genes of the cancerous region in each group.

Term	ID	Nano-DEN				Nano-DEN + HFD			
		Gene Ratio	Related Targets	*p* Value	Rank	Gene Ratio	Related Targets	*p* Value	Rank
Cell cycle	mmu04110	19/243	Atp5g2/Sdhd/Mgst3/ Mgst1/Rac1/Nd6/ Nd5/Nd4 l/Nd4/ Nd3/Nd2/Nd1/Cytb/ Cox3/Cox2/Cox1/ Atp8/Atp6/Cox5a	9.12 × 10^−6^	61	7/243	Chek2/Pkmyt1/ Ywhaq/Cdc14b/Plk1/ Cdkn1b/Cdk6	5.38 × 10^−2^	25 ↑
TGF-β signaling pathway	mmu04350	5/243	Cdc23/Ywhaq/ Cdc14b/Cdkn2d/ Cdc25a	2.53 × 10^−1^	131	5/243	Ppp2r1b/Tgfbr1/ Smad6/Bmp8b/Amh	1.16 × 10^−1^	61 ↑
HIF-1 signaling pathway	mmu04066	4/243	Prickle4/Serpinf1/ Rac1/Camk2b	8.95 × 10^−2^	98	5/243	Pgk1/Insr/Cdkn1b/ Camk2 g/Camk2b	1.98 × 10^−1^	75 ↑
Wnt signaling pathway	mmu04310	4/243	Pdk1/Gapdh/ Eno3/Camk2b	3.75 × 10^−1^	85	5/243	Lgr6/Map3k7/Wisp1/ Camk2 g/Camk2b	4.88 × 10^−1^	86
IL-17 signaling pathway	mmu04657	4/243	Echs1/Fads1/ Hadhb/Scd1	8.76 × 10^−2^	187	4/243	Mapk15/Map3k7/ Ikbkg/Elavl1	2.34 × 10^−1^	94 ↑
Chemical carcinogenesis-reactive oxygen species	mmu05208	3/243	Pla2g4f/Rac1/Pxn	2.09 × 10^−1^	12	4/243	Ndufa5/Ikbkg/ Hgf/Gm3776	8.59 × 10^−1^	98
Fatty acid metabolism	mmu01212	3/243	Tgfbr1/Nbl1/Inhba	4.80 × 10^−1^	83	4/243	Elovl7/Ehhadh/ Hsd17b12/Elovl2	8.76 × 10^−2^	126
VEGF signaling pathway	mmu04370	2/243	Scd1/Cd36	7.01 × 10^−1^	116	3/243	Sphk2/Sh2d2a/Pla2g4a	2.09 × 10^−1^	130
PPAR signaling pathway	mmu03320	2/243	Mapk6/Cxcl2	7.13 × 10^−1^	185	3/243	Ehhadh/Plin4/Fabp5	4.37 × 10^−1^	151 ↑
TNF signaling pathway	mmu04668	1/243	Cxcl2	9.56 × 10^−1^	262	2/243	Map3k7/Ikbkg	8.17 × 10^−1^	220 ↑

The symbol ↑ highlighted in red indicates an increase in ranking position.

## Data Availability

The data used to support the findings of this study are available from the corresponding author upon reasonable request.

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
