# Peer review of "Promotion Effect of Coexposure to a High-Fat Diet and Nano-Diethylnitrosamine on the Progression of Fatty Liver Malignant Transformation into Liver Cancer"

_ijms, 2023, doi:10.3390/ijms241814162_

Round 1

Reviewer 1 Report (New Reviewer)

Comments and Suggestions for Authors

Congratulations to the authors for the quality of their work.

- Please, in the references section you must complete the information of the number [15].

- Lines 95 to 96: “…and peroxisome proliferator-γ (peroxisome proliferator-α)”.

The authors indicate with two different Greek letters. Correct the defect.

- Lines 130 to 133: “The results of tumor incidence at the 25th weekshowed that 41% of the total number of mice developed tumors from high-fat group (C+ group), 32% of the total number of mice developed tumors from nano-DEN group (M group). Importantly, the tumor incidence at the 25th week in the coexposure group (M+ group) was greater than 78%”.

The nomenclatures do not coincide with the figure caption (Figure 1). They appear changed in the text.

- Lines 140 to 144: “Compared with the control group, the steatosis scores were increased in the high-fat group (3.80 ± 0.35 of control, p< 0.001) and the nano-DEN group (3.20 ± 0.45 of control, p< 0.01). Surprisingly, the degree of steatosis was also increased in the coexposure group (4.00 ± 0.35 of control,p< 0.001), and it was the most severe compared with the other two groups”.

Please correct the values for each group. They do not correspond to figure 1B "steatosis scores".

- Lines 160 to 161: “Although there was no significant difference in serum total protein or total cholesterol content, these two contents showed an increasing trend after stimulation”.

Quite the opposite for protein levels, or similar to high-fat diets for TCHO levels. They have decreasing trends.

- Lines 253 to 256: “The KEGG results showed that the DEGs induced by nano-DEN were enriched in glycerophospholipid metabolism, ether lipid metabolism, glycosphingolipid biosynthesis lacto and neolacto series, fat digestion and absorption, and the cholesterol metabolism pathway (Fig. 4B). The coexposure groups were mainly enriched in the AMPK signaling pathway, FoxO signaling pathway”.

It seems to me that in figures 4B and 4C they seem to say the opposite, they are interpreted as reductions in the expressions.

- Line 428. 20%-30% of energy as fat are common patterns of high-fat diets. However, the authors provide higher quantities (60%). You must specify in this section that your objective is focused on obtaining a specific fatty liver model. Have the authors verified whether the results are similar by reducing the percentage of fat supplied in the diet over long periods of time?

- On the other hand, much literature talks about the benefits of the ketogenic diet by increasing the fat content, and here there is no apparent benefit. As a suggestion, I encourage the authors to add a brief paragraph in the discussion about the impact of the ketogenic diet and whether it is a comparable model to your high-fat diet model.

Author Response

Point-by-point reply to Reviewer #1: 

We would like to express our heartfelt gratitude for reviewer’s valuable feedback and recognition of our research work. We are willing to accept and incorporate the improvements you have proposed. With your assistance, our manuscript is more refined and professional. The replies to the comments are in the order in which the reviewer is made.

  1. Please, in the references section you must complete the information of the number [15].

Response: Thank you for your thorough review. Regarding article number 15, titled "p16," perhaps the title name of this article seems to be incomplete information because of the brevity of the title name of this article. Here is the detailed citation information for this article: Serra S, Chetty R. p16. J Clin Pathol. 2018 Oct;71(10):853-858. doi: 10.1136/jclinpath-2018-205216. Epub 2018 Aug 3. PMID: 30076191.

  1. Lines 95 to 96: “…and peroxisome proliferator-γ (peroxisome proliferator-α)”.The authors indicate with two different Greek letters. Correct the defect.

Response: Thanks for your meticulous review and for pointing out our oversight. We apologize for our negligence in incorrectly inputting γ in our work. Based on your feedback, we have made the corrections to reflect the accurate information: PPAR-γ, as follows:

“peroxisome proliferator-γ (PPAR-γ).” (Revised version: page 5, line 98)

  1. Lines 130 to 133: “The results of tumor incidence at the 25th week showed that 41% of the total number of mice developed tumors from high-fat group (C+ group), 32% of the total number of mice developed tumors from nano-DEN group (M group). Importantly, the tumor incidence at the 25th week in the coexposure group (M+ group) was greater than 78%”.

The nomenclatures do not coincide with the figure caption (Figure 1). They appear changed in the text.

Response: Thank you for your careful attention to detail and for pointing out our oversight. We apologize for our negligence. Based on the issues you raised, we have made the necessary corrections to the figure captions and updated them to the correct versions: 

“high-fat group (C+), nano-DEN group (M)” (Revised version: page 7, line 149)

  1. Lines 140 to 144: “Compared with the control group, the steatosis scores were increased in the high-fat group (3.80 ± 0.35 of control, p< 0.001) and the nano-DEN group (3.20 ± 0.45 of control, p< 0.01). Surprisingly, the degree of steatosis was also increased in the coexposure group (4.00 ± 0.35 of control,p< 0.001), and it was the most severe compared with the other two groups”.

Please correct the values for each group. They do not correspond to figure 1B "steatosis scores".

Response: Thank you for your diligence and careful review, pointing it out. We have re-evaluated our data and made the necessary corrections accordingly. We would like to apologize for not including the updated version of our steatosis scores criteria in our previous submission. Please find the updated data below:

 “Compared with the control group, the steatosis scores were increased in the high-fat group (2.40 ± 0.20 of control, p< 0.001) and the nano-DEN group (1.80 ± 0.37 of control, p< 0.01). Surprisingly, the degree of steatosis was also increased in the coexposure group (2.60 ± 0.24 of control, p< 0.001)” (Revised version: page 6, lines 144-146)

  1. Lines 160 to 161: “Although there was no significant difference in serum total protein or total cholesterol content, these two contents showed an increasing trend after stimulation”.

Quite the opposite for protein levels, or similar to high-fat diets for TCHO levels. They have decreasing trends.

Response: We apologize for the oversight in our writing, and we appreciate you bringing it to our attention. Your point is very interesting, and we conducted extensive research in medical literature. We found that when the liver experiences diseases such as cirrhosis or liver cancer, protein synthesis function is impaired, leading to a gradual decrease in the level of total protein (TP) in the serum. However, the difference is not significant. Conversely, excessive intake of dietary fats (high-fat diet) can increase the level of total cholesterol (TCHO) in the serum, commonly observed in cardiovascular diseases caused by atherosclerosis and hyperlipidemia. However, in severe liver diseases such as cirrhosis, anemia, malnutrition, and malignant tumors, TCHO may actually decrease. What you point out is very important, and it indirectly explains our data. Therefore, we have made the necessary revisions to our manuscript as follows:

 “Although there were no significant differences in serum total protein (TP) and total cholesterol (TCHO), both exhibited distinct trends after stimulation. The TP levels gradually decreased, whereas TCHO levels showed a correlation with high-fat diet consumption, leading to an increase. However, it is noteworthy that liver damage induced by nano-DEN actually resulted in a decrease in TCHO levels.” (Revised version: page 8, lines 162-165)

  1. Lines 253 to 256: “The KEGG results showed that the DEGs induced by nano-DEN were enriched in glycerophospholipid metabolism, ether lipid metabolism, glycosphingolipid biosynthesis lacto and neolacto series, fat digestion and absorption, and the cholesterol metabolism pathway (Fig. 4B). The coexposure groups were mainly enriched in the AMPK signaling pathway, FoxO signaling pathway…”.

It seems to me that in figures 4B and 4C they seem to say the opposite, they are interpreted as reductions in the expressions.

Response: We appreciate you raising this issue as it gives us an opportunity to clarify and improve our communication!

In Fig.4B, the arrows labeled in front of the pathways represent the expression levels of the relevant genes on the transcriptional level (which can be understood as either downregulation or upregulation). The ranking is from top to bottom, and the size of the circles represents the degree of enrichment (which can be understood as the strength of correlation or activity).

These results can also be seen as screening for the pathways most affected by coexposure. These data need to be analyzed in conjunction with Table 3. Taking the lipid metabolism pathway in Fig.4B as an example, four pathways related to lipid metabolism are significantly downregulated. This indicates that coexposure of a high-fat diet and nano-DEN makes lipid metabolism more difficult compared to the sole intake of nano-DEN, thus exacerbating liver damage.

Taking Fig.4C as another example, NOD promotes inflammation and inhibits cancer gene expression, showing an upregulation. AMPK and FoxO inhibit cancer-related responses, also showing an upregulation. The upregulation of pro-inflammatory factors combined with the downregulation of anti-inflammatory factors indicates that coexposure exacerbates inflammatory responses, thereby promoting liver carcinogenesis and worsening liver damage.

Therefore, both the results in Fig. 4B and Fig. 4C prove that coexposure alters the microenvironment of the liver, accelerating and exacerbating carcinogenesis.

To aid in reader comprehension, we have added additional data and explanations to further support our findings. We sincerely appreciate you raising this question, as it allows us to address any ambiguities and improve the clarity of our work:

“These pathways play a crucial role in lipid metabolism reactions. Analysis of the gene ratios in Table 3 showed that the genes in the first four pathways mentioned above were significantly downregulated at the transcriptional level, while the genes in the Ether lipid metabolism pathway were significantly upregulated (p < 0.05). Moreover, as shown in Fig.4C, the DEGs between the cancerous and paracancerous regions of liver tissues from mice of the coexposure group were mainly enriched in the NOD-like receptor signaling pathway, FoxO signaling pathway, and AMPK signaling pathway, which are closely associated with inflammatory responses. Additionally, based on Table 3, it was observed that 62.5% (5/8) of the differentially expressed genes related to the NOD-like receptor signaling pathway, which promotes inflammatory responses, showed high expression at the transcriptional level, while both the AMPK signaling pathway and the FoxO signaling pathway, which inhibit inflammatory responses, exhibited low expression in 80% (4/5) of the differentially expressed genes (p < 0.05). These results suggest that coexposure exacerbates the up-regulation of inflammation-related genes in mouse liver tissue, promoting the malignant transformation of fatty liver into cancer.” (Revised version: page 13, lines 256-268)

“The above results suggest that coexposure aggravated the high expression of genes related to inflammation, hypoxia, and aging pathways in the mouse liver, promoting the malignant transformation of fatty liver into cancer.” (Revised version: page 14, lines 272-275)

  1. Line 428. 20%-30% of energy as fat are common patterns of high-fat diets. However, the authors provide higher quantities (60%). You must specify in this section that your objective is focused on obtaining a specific fatty liver model. Have the authors verified whether the results are similar by reducing the percentage of fat supplied in the diet over long periods of time?

Response: Thank you for your professional suggestions. In this experiment, we used the Diet-Induced Obesity (DIO) series diets to feed mice, which are widely employed for the construction of diet-induced obesity models in both mice and rats. The aim was to rapidly simulate the pathological and physiological changes, such as metabolic disorders, that occur in mice after consuming an unhealthy diet. This approach allows for the development of a more stable diseased state compared to the conventional high-fat model, thus better mimicking the occurrence and progression of diet-related diseases in humans.

The proportion of energy derived from high-fat diets can vary based on individual needs and goals. Generally, high-fat diets predominantly consist of fat, with relatively lower intake of carbohydrates and proteins. The following is an example of recommended energy distribution for a typical high-fat diet (note: There is a strong correlation between this and the subsequent question you raised about distinguishing between high-fat diets and ketogenic diets):

â‘ Fat: Approximately 60-75% of total energy intake.

â‘¡Carbohydrates: Approximately 5-20% of total energy intake.

â‘¢Protein: Approximately 10-30% of total energy intake.

The formulation of the diet we used aimed to simulate the standard high-fat diet-induced obesity model and consisted of 60% fat, 20% carbohydrates, and 20% protein in terms of total energy intake. We chose this diet to rapidly simulate the effects of a high-fat diet and observe a clearly discernible disease state.

Furthermore, although we did not investigate the effects of reducing fat content in this experiment, in other studies, the Western Diet (WD) method, which involves feeding a diet containing 45% fat as a proportion of energy, has been utilized for modeling purposes. The results obtained from that study were similar to those observed in our high-fat group, albeit with a prolonged onset of noticeable liver disease. In future experiments, we will take into consideration the reviewer's suggestion and employ different high-fat diets to account for a more comprehensive analysis. We appreciate the reviewer's insightful comments, as they have contributed to improving the rigor of our manuscript.

According to the reviewer's suggestion, we have rewritten this section to add more detail as follows:

 “In addition, the control group and nano-DEN group were provided with regular feed, while the high-fat group and co-exposure group were given obesity model series diets (DIO series Diets). The regular feed had a fat energy supply rate of 10%, whereas the high-fat diet had a fat energy supply rate of 60%. Both the regular feed and high-fat feed were sourced from Beijing Huafukang Biotechnology Co., Ltd. (Beijing, China). The composition of the high-fat diet for mice (No. H10060) consisted of casein, dextrin, sucrose, cellulose, soybean oil, lard, various minerals, various vitamins, and choline, with fat accounting for 60% of the total energy intake, carbohydrates accounting for 20%, and protein accounting for 20%.” (Revised version: pages 22, lines 440-447)

  1. On the other hand, much literature talks about the benefits of the ketogenic diet by increasing the fat content, and here there is no apparent benefit. As a suggestion, I encourage the authors to add a brief paragraph in the discussion about the impact of the ketogenic diet and whether it is a comparable model to your high-fat diet model.

Response: Thank you very much for your professional guidance and constructive feedback. Although both high-fat and ketogenic diets are characterized by high-fat content, there are differences between the two in terms of the degree of carbohydrate restriction.

High-fat diets (HFD) emphasize high-fat intake without restricting carbohydrate intake, while a ketogenic diet is a low-carbohydrate, moderate-protein, and high-fat dietary pattern. The main principle of a ketogenic diet is to limit carbohydrate intake, forcing the body into a state of ketosis, where fats are used as the primary source of energy.

In HFD, fat energy supply ranges typically from 40% to 60%, while in ketogenic diets, fat energy supply needs to reach 80%, 90%, or even 95%. Carbohydrates and proteins in high-fat diets usually account for over 20%, whereas carbohydrates in a ketogenic diet generally comprise around 5% of total calorie intake, and proteins account for less than 15%.

Regarding their health effects, the effectiveness and suitability of ketogenic and high-fat diets vary among individuals, and it is not simply a matter of whether they are healthy or not. To address this, we consulted experts and doctors at hospitals.

Some advantages of a ketogenic diet include potential weight and waist reduction, which may be more effective for certain groups such as obese individuals and those with type 2 diabetes; it may improve cholesterol levels, particularly by increasing HDL (good cholesterol) levels; and it may have protective effects against certain neurodegenerative diseases, such as epilepsy and Parkinson's disease. However, there are also potential health risks and limitations, such as a lack of dietary fiber, which may lead to constipation and gastrointestinal issues, as well as the possibility of nutritional imbalances, including deficiencies in certain vitamins, minerals, and other essential nutrients. It is not suitable for certain individuals, such as those with liver disease, pancreatitis, or gallbladder disease. However, further research is needed to investigate the potential long-term effects and unknown risks associated with a ketogenic diet.

In conclusion, I believe that every dietary pattern has its pros and cons, and the key lies in striking a balance rather than advocating or completely avoiding a particular approach. Your suggestions are interesting and innovative for improving our future research, and I appreciate your thought-provoking ideas again.

With your suggestion, we have added the content in the discussion as followed:

“In recent years, the consumption of high-fat ketogenic diets has gained increasing attention from many individuals, particularly those struggling with obesity, as a means to control cholesterol levels and achieve weight loss. However, there is much controversy surrounding the potential risks and effects of this approach.” (Revised version: page 3 lines 51-54)

“Taking the recent popular ketogenic diet as an example, it is a high-fat dietary habit. This diet may have some benefits for certain populations such as individuals seeking continued weight loss, type 2 diabetes patients, and those with neurodegenerative diseases (such as epilepsy and Parkinson's disease). However, it may not be suitable for individuals with liver disease, pancreatitis, or gallbladder disease. Furthermore, the ketogenic diet imposes strict requirements on carbohydrate intake, which must be maintained within a very limited range. Failure to properly control carbohydrate intake may increase the burden on the liver and worsen liver damage. Additionally, if individuals following a high-fat diet, including those on the ketogenic diet, also have a habit of consuming foods with high levels of nitrites, such as Korean kimchi, salted fish, bacon, and other preserved products, this undoubtedly accelerates and exacerbates the formation of fatty liver disease and its progression to cancer. We believe that every dietary approach has pros and cons, and the key lies in balancing and choosing wisely. It is essential to consult with professionals and select a diet that suits one's individual needs.” (Revised version: page 17, lines 308-319)

Your professional insights and suggestions are truly invaluable, whether it is ensuring grammatical accuracy or improving the clarity and comprehensibility of our manuscript.

We sincerely appreciate this invaluable opportunity you have given us!

Reviewer 2 Report (New Reviewer)

Comments and Suggestions for Authors

Yin et al. established a liver injury model of fatty liver malignant transformation into liver cancer by continuously feeding mice a high-fat diet to simulate people's daily high-fat diet habits and explored the effect and mechanism of coexposure to high-fat and the chemical poison nano-DEN on liver cancer in mice. The results are generally well presented, with statistical analysis appropriately applied. The discussion integrates the results with existing literature, and potential mechanisms are proposed.

Below are my comments:

English needs to be polished.

Rationale is not well introduced. The authors wrote the rationale in the Discussion section.

Why chose these 2 insults? What is the estimated prevalence of these 2 insults occur simultaneously in the general population?

The nomenclatures of animal grouping are weird. Why C+, M, M+ were used?

Table 1: WQhat are the symbols meaning?

Fig.2.: Only A and B were seen, C, D, E, and F are missing.

Fig.6 was not mentioned in the text.

Table 2: What are # and ## stand for?

Fig.4: Why HFD group was not compared?

Fig.5: Why HFD group was not compared?

Statistics: Why HFD group was not compared?

Why the authors only compared 2 hit and nano-DEN only?

There are plenty of careless editing errors that need to be corrected. Examples are Examples are as following: showthatthe (line 41), coexposuresignificantly (line 41), etc...

Inadequate yellow highlights in the text were seen.

Acronyms should be used correctly. Examples are nano-DEN (line 110) and etc...; high-fat diet (HFD);

Overall, extensive proof-reading and text editing are needed.

Comments on the Quality of English Language

English quality must be improved.

Author Response

Reply to Reviewer #2: 

We appreciated the reviewer’s effort of improving our manuscript. We have considered each of the suggestions in turn and have rewritten the manuscript accordingly to the reviewer’s comments.

Additionally, we noticed that some of the questions raised by the reviewer belong to the same category, so we have addressed them together point by point, ensuring that none of them were overlooked.

  1. English needs to be polished.
  2. Inadequate yellow highlights in the text were seen. Acronyms should be used correctly. Examples are nano-DEN (line 110) and etc...; high-fat diet (HFD); Overall, extensive proof-reading and text editing are needed. English quality must be improved.

Response: We apologize for the oversight in our writing, and we appreciate you bringing it to our attention. It has been brought to our attention that there are instances where missing spaces have resulted in words being connected together, affecting readability. After conducting an internal investigation, we found that this issue occurred due to an outdated version of our computer software, which failed to detect these errors during the file conversion process. Our last manuscript has undergone thorough editing by two experienced American teachers with strong academic backgrounds in scientific research. Their expertise ensures linguistic accuracy and overall content quality. In fact, reviewers expressed satisfaction with the revised manuscript in the previous submission. Therefore, it is possible that the initial version we uploaded unintentionally misled you, creating the impression of language-related issues.

To enhance the readability of the document, we have once again revised the text, including using yellow highlights to emphasize key content. Additionally, we have sought the assistance of native English-speaking experts to proofread and refine the revised manuscript. To differentiate the previous modifications, we will now highlight them in blue instead of yellow.

To effectively address this matter, we have extensively proofread and edited our manuscript. We have uploaded corrected versions in both Word and PDF formats. The Word document contains revised content with appropriate spacing reinstated, while the high-resolution PDF ensures optimal clarity and legibility.

Once again, we sincerely apologize for any inconvenience caused by our oversight and any confusion it may have caused during the review process. We genuinely appreciate your understanding and continued support!

  1. Rationale is not well introduced. The authors wrote the rationale in the Discussion section.

Response: Thank you very much for your professional guidance and constructive feedback.

The questions raised by the reviewer2 in Q4 were thoughtful and logical, providing us with valuable insights. Based on the reviewer's questions, we have rewritten the introduction section, attempting to address the rationale from the following perspectives:

①Introducing the widespread presence of high-fat diet and nitrosamine&Emphasizing the attention given to high-fat diet and ketogenic diet:

“high-fat diets (HFD) and nitrosamines are both common factors in daily diets. In recent years, the consumption of high-fat ketogenic diets has gained increasing attention from many individuals, particularly those struggling with obesity, as a means to control cholesterol levels and achieve weight loss. However, there is much controversy surrounding the potential risks and effects of this approach.”

â‘¡: Discussing the differences in dietary habits among countries and regions:

“Dietary habits vary across countries, impacting the consumption and exposure to HFD and nitrosamines. For instance, China's cuisine features fatty dishes like stir-fries and deep-fried snacks, resulting in a high intake of fats and moderate exposure to nitrosamines from preserved foods. Similarly, South Korea's traditional dishes such as samgyeopsal and bibimbap have a higher fat content, leading to moderate consumption of fats and nitrosamines. In contrast, the Inuit people in the Arctic follow an HFD primarily from marine mammals, but their remote location and traditional hunting practices limit their exposure to nitrosamines. It is intriguing to note that despite these protective factors, these countries still demonstrate an elevated susceptibility to fatty liver and cancer, as evidenced by the liver cancer incidence rates ranking provided by the International Agency for Research on Cancer (IARC/WHO).”

â‘¢Highlighting the importance and objectives of the research:

“Therefore, conducting a comprehensive study on the simultaneous existence of HFD and nitrosamines is crucial for improving our understanding of cancer development mechanisms, devising effective prevention strategies, and promoting public health outcomes.”

â‘£Exploring the potential impact of dual factors on liver cancer incidence:

“Recent studies have reported that HFD and exposure to chemical toxic substances, which are key factors in the development of fatty liver disease, can induce obesity and insulin resistance. This results in the excessive lipids accumulation in the liver, creating an inflammatory microenvironment and ultimately promoting the progression from fatty liver disease to malignant liver cancer [4]. Long-term consumption of an HFD disrupts liver metabolism and promotes lipid accumulation, resulting in the phosphorylation of insulin signaling transcription pathways, insulin resistance, and altered expression of Akt, GLUT4, and lipogenic genes [5, 6]. Insulin resistance triggers the breakdown of hepatic fat, with free fatty acids generated during this breakdown stimulating the expression of factors involved in cholesterol synthesis and fatty acid synthesis [7, 8]. The accumulation of metabolized lipid substances and chemical toxins can cause lipid peroxidation, damage the cell membrane's permeability, and lead to mitochondrial damage and inflammation, which further impairs the liver's normal metabolism [9].” (Revised version: pages 3-4, lines 51-76)

  1. Why chose these 2 insults? What is the estimated prevalence of these 2 insults occur simultaneously in the general population?

Response: Thank you for the interesting question. The high-fat diet and nitrosamines are commonly present in daily diet, and the intake of high-fat ketogenic diet is increasingly becoming a concern for many people. Therefore, we decided to study the insult caused by these two factors. For example, different dietary habits in each country have an impact on the incidence of liver cancer. In Chinese cuisine, preserved foods such as pickled vegetables and meats contain a large amount of nitrosamines. However, there are also regional differences in China, with northern people eating more preserved foods than those from southern regions. Kimchi is ubiquitous in Korea and is a staple food in every meal, which may lead Korean people to consume more nitrites than people from other countries. Eskimos living in the Arctic Circle and Western countries generally have a high-fat diet habit compared to Asian countries. Interestingly, the probability of developing fatty liver and cancer in these countries is very high, and the incidence rate of liver cancer in the world listed by WTO is generally ranked top.

Many people are likely to be exposed to both factors in their diet. Therefore, studying their simultaneous existence is of great significance for a better understanding of the mechanism of cancer development, devising prevention strategies, and improving public health.

As for estimating the probability of developing cancer when both stimuli exist simultaneously, it is a complex problem that requires consideration of multiple factors. The intake and exposure level of high-fat diets and nitrosamines are influenced by various factors such as demographic, geographic, and cultural factors, making the estimations different. Taking China as an example and based on consultations with epidemiology professionals and literature review, we estimate that in China, the incidence of liver cancer caused by both insults simultaneously accounts for 40% of the total liver cancer population. According to WTO statistics, the incidence rate of liver cancer in the general population of China is 0.02%. Therefore, we estimate that the probability of both insults occurring simultaneously in the general population of China is approximately 0.8%.

  1. The nomenclatures of animal grouping are weird. Why C+, M, M+ were used?

Response: Thank you for bringing that up. Due to the large number of images and tables, we have decided to use abbreviations to simplify and optimize the layout. In order to maximize the visual clarity and conciseness of the images and tables, we have minimized the length of the names in the graphics.

Our approach involves using "C" to represent the control group, which refers to the normal mice being fed a high-fat diet without any toxic substances. It is worth noting that high-fat diets are commonly observed in daily life among individuals such as athletes and those following a ketogenic diet. At times, this group can be considered as the control group, hence denoted as "C+".

On the other hand, for the experimental group receiving nano-DEN, a toxic substance, we use the abbreviation "m" to represent the model group. By combining the previous "M" with the addition of a high-fat diet symbolized by "+", we denote the group as "M+". This indicates that the mice in the model group, already receiving nano-DEN, are further fed a high-fat diet.

  1. Table 1: What are the symbols meaning?
  2. Table 2: What are # and ## stand for?

Response: We would like to sincerely apologize for the oversight in our manuscript regarding the formatting error! We inadvertently misplaced the symbol explanations from Table 1 and included them below Table 2.

We have carefully reviewed the manuscript and made the necessary corrections to rectify this mistake. The symbol explanations have now been correctly placed beneath Table (Revised version: page 8, lines 167-168), ensuring that the tables and their corresponding descriptions align properly.

Thank you for bringing this error to our attention, and we apologize for any confusion it may have caused. We greatly appreciate your thorough review of our manuscript and the opportunity to improve its overall quality.

  1. 2.: Only A and B were seen, C, D, E, and F are missing.

Response: I’m so sorry for the oversight in Figure 2 of our manuscript. We unintentionally added the labeling of each panel in the image, representing A, B, C, D, E, and F, but actually, we don’t need these letters. We sincerely regret this mistake and any confusion it may have caused.

We have rectified this error by deleting the corresponding letters to each panel in the revised version of the annotation in our manuscript. (Revised version: pages 17-18, lines 228-229)

Thank you for bringing this to our attention!

  1. 6 was not mentioned in the text.

Response: I apologize for any confusion, but we did mention Figure 6 in the text and provided relevant discussion. It might have been overlooked due to formatting issues as this particular page with Figure 6 was separate from the rest of the text. To facilitate the review process, we put there as followed:

“To visualize and compare the differences in mice, we generated a schematic diagram that ranks the activation status of selected signaling pathways in our experiments, incorporating both previous findings and our current results (Figure 6). Previous experiments observed that continuous feeding of mice with an HFD resulted in significant liver damage, including steatosis, in 41% of the mice at 25 weeks, fibrosis and cirrhosis in 61% of the mice at 35 weeks, and hepatocellular carcinoma in 89% of the mice at 52 weeks [30].  Consistent with our previous studies, when mice were solely exposed to nano-DEN, 61% of the mice exhibited evident pathological signs of cancer at 25 weeks, and 62.5% of the mice developed liver cancer at 35 weeks [23]. To enable comparison, we replicated their previous dosing and divided the mice into three groups, administering different treatments while conducting RNA sequencing at 25 weeks.  Notably, when mice were coexposed to an HFD and nano-DEN, the time for the development of liver cancer was significantly shortened to 25 weeks compared to the other two groups. Through gene sequencing of liver tissues from three groups of mice, we made an interesting discovery. The consumption of an HFD significantly enhanced the activation of oncogenic signaling pathways, with particular emphasis on the NAFLD and chemical carcinogenesis-ROS, which ranked prominently. Intriguingly, the ranking of cancer suppressor-related pathways such as the p53, FoxO, and AMPK pathways exhibited a significant decrease in comparison. Moreover, upon exposure to nano-DEN, we observed the activation of pathways associated with cancer inhibition. Tumor suppressor genes were found to be upregulated, consequently activating pertinent pathways including FoxO, AMPK, and p53. Remarkably, when mice were coexposed to both nano-DEN and an HFD, the activity of carcinogenesis-related signaling pathways increased significantly, surpassing the effects seen in the individual treating groups. In comparison to the nano-DEN group, the coexposure group exhibited a greater suppression in cancer suppressor-related pathways. These findings shed light on the potential implications. It appears that an HFD, particularly when combined with the exposure to nano-DEN, may compromise the body's natural self-repair mechanisms and its ability to inhibit cancer development. This phenomenon can be attributed to the suppression of cancer suppressor-related pathways, resulting in decreased activity of genes involved in these pathways. Consequently, the body's response to carcinogens is accelerated, potentially exacerbating the risk of cancer. In present work, we focused on critical proteins related to cancer development and progression and identified three representative protein pathways associated with the signaling pathways. Through subsequent experimental analysis, we examined the expression patterns, interactions, and functional roles of these key proteins. Our results enhance the understanding of the complex molecular mechanisms underlying the observed phenomena and strengthen the validity of our conclusions. By utilizing a range of techniques such as HE staining, immunohistochemical staining, and quantitative analysis using the Nuance multispectral imaging system, we were able to precisely identify and quantitatively analyze the alterations occurring within these pathways. This comprehensive approach bolstered the support for our conclusions.” (Revised version: pages 17-18, lines 323-356)

  1. 4, Fig.5, and Statistics: Why HFD group was not compared?
  2. Why HFD group was not compared?
  3. Why the authors only compared 2 hit and nano-DEN only?

Response: We appreciate you raising this issue as it gives us an opportunity to clarify and improve our communication.

Here are the reasons why we did not include the HFD (high-fat diet) group in our study:

In the section depicted in Fig.3, we conducted a cross-comparison of three groups and identified 54 differentially expressed genes that were shared among all three groups. Through these genes, we were able to identify three enriched pathways that play crucial roles in carcinogenesis. Our objective was to demonstrate the upregulation of cancer-related genes induced by these factors by measuring the mRNA expression levels of these pathways in the liver tissues of mice from the three groups.

For the experiment illustrated in Fig.3, we obtained liver tissues, while for Fig.4, we collected liver cancer tissues and adjacent non-cancerous tissues from the mice.

In this chapter, our primary focus was comparing the cancerous regions with the adjacent non-cancerous regions. However, the mice that were solely on a high-fat diet did not exhibit a significantly higher incidence of cancer. Instead, they primarily developed fatty liver and experienced more inflammatory infiltration compared to the control group. Therefore, analyzing the gene expression in the cancerous and non-cancerous regions of the high-fat diet group would not yield realistic results.

Furthermore, since the critical factor promoting carcinogenesis was the intake of nano-DEN from daily dietary consumption of pickles, salted fish, and bacon, the high-fat diet was not the main focus of our comparisons but rather served as a foundation.

Each section of our study builds upon the previous one, following a logical progression that gradually narrows down our focus to specific key points. Maybe our description was too brief and caused confusion. To aid in reader comprehension, we have added additional content. We sincerely appreciate you raising this question, as it allows us to address any ambiguities and improve the clarity of our work:

“We obtained liver tissues from mice in the HFD group, nano-DEN group, and coexposure group, in order to identify the enriched pathways that play a key role in the process of carcinogenesis.” (Revised version: page 11, lines 223-224)

“Next, we collected liver cancer tissues from the mice in the nano-DEN group and the coexposure group. To focus on the changes occurring from precancerous to fully developed cancer stages, we compared the gene expression levels between the cancerous and paracancerous regions of liver tissues from mice of the coexposure group.” (Revised version: page 13, lines 250-253)

Thank you for your time and valuable input!

All the aspects mentioned have been meticulously considered and revised. With your patient guidance, our article has gradually been refined into a more polished piece. Thank you for patiently guiding me. I can sense your guidance and patient goodwill from the questions you have raised. Your meticulous review, professional advice, clear insights, and point-by-point suggestions have greatly enhanced the rigor of our article. As we watched our manuscript gradually grow and become enriched, we felt a great sense of fulfillment. We sincerely thank you for your dedicated help!

Round 2

Reviewer 2 Report (New Reviewer)

Comments and Suggestions for Authors

The authors have addressed all my concerns.

This manuscript is a resubmission of an earlier submission. The following is a list of the peer review reports and author responses from that submission.

Round 1

Reviewer 1 Report

Comments and Suggestions for Authors

In the present manuscript, Li et al. studied the impact of the simultaneous exposure to a high-fat diet and to nano-diethylnitrosamine in the progression of nonalcoholic fatty liver disease to liver cancer, in mice. In comparison with the single exposure to a high-fat diet or to nano-diethylnitrosamine, coexposure doubled tumor incidence, despite bringing minor increases to the inflammation and steatosis indices. A transcriptomic analysis identified differentially expressed genes in the coexposure group, which were related to inflammation, hypoxia, and aging. Coexposure also up-regulated four proteins in the tumor microenvironment: COX-2, β-catenin, HIF-1α, and p16, whose increased levels have previously been related to the incidence of liver cancer.

The manuscript is clearly written and contributes insight into the pathophysiology of the progression of nonalcoholic fatty liver disease to liver cancer.

Minor issues that should be addressed:

1. In section 2.1, please start by briefly summarizing the experiment whose results are here described. This summary will be helpful to the readers of this paper, since the results are described before the materials and methods.

2. In the legend of Figure 1A, please indicate the ruler units.

3. In Figure 1B, right the graph is labeled "Pimelosis scores". Please correct to "Steatosis scores", if adequate.

4. Under Table 1, please include a list of the abbreviation used in this table.

5. Figures S1-S4 are a bit hard to read, please make sure to provide high-resolution images in the revised version of the manuscript.

6. While using "KEGG" for the first time, please include its full name.

7. In the title of section 2.7, please replace "aggravated" with a more specific word, such as "increased".

8. In the discussion, while summarizing the major findings of this study, please also mention the corresponding figures/figure panels.

9. In line 364, please include the full name of "SPF".

Comments on the Quality of English Language

English quality is good. In lines 151-152, please correct "coexposuregroup" to "coexposure group". In lines 158-162, it is suggested to split and reorganize this long sentence into shorter ones.

Author Response

Point-by-point reply to Reviewer #1: 

We would like to express our heartfelt gratitude for reviewer’s valuable feedback and recognition of our research work. We are willing to accept and incorporate the improvements you have proposed. With your assistance, our manuscript is more refined and professional. The replies to the comments are in the order in which the reviewer is made.

  1. In section 2.1, please start by briefly summarizing the experiment whose results are here described. This summary will be helpful to the readers of this paper, since the results are described before the materials and methods. 

Response: Thank you very much for your constructive and professional suggestions. As the reviewer suggested, we have added a brief summary of the experimental as follows:

“This section aimed to investigate cancer progression through visual observation, employing the hematoxylin and eosin (H&E) staining method. Both macroscopic anatomical changes in the liver and microscopic pathological alterations in tissue sections were examined. Quantitative analysis was then conducted to evaluate the incidence and severity of liver tissue damage, which was scored based on three criteria: the number of individual tumor occurrences, the inflammatory score of tumors, and hepatic steatosis (fat accumulation) score.” (Revised version: page 5, lines 101-106)

  1. In the legend of Figure 1A, please indicate the ruler units.

Response: Thank you immensely for astutely noting this particular point. A common steel ruler with units in centimeters was used for scale in the image depicting the liver tissue. With your suggestion, we have updated the legend of Figure 1 in the revision, and added the ruler units as follows: “(with a standard steel ruler in centimeters)” (Revised version: page 6, lines 132)

  1. In Figure 1B, right the graph is labeled "Pimelosis scores". Please correct to "Steatosis scores", if adequate.

Response: Thank you very much for your diligent clarification. We have followed your professional advice and updated the graph in Figure 1B. (Revised version: page 6,  Figure 1-amended)

  1. Under Table 1, please include a list of the abbreviation used in this table.

Response: Thank you for your careful reading and professionalism in your suggestion. 

As the reviewer suggested, we have added a list of the abbreviation used in table 1 as follows: “Abbreviations: GOT, glutamic oxaloacetic transaminase; GPT, glutamic pyruvic transaminase; ALP, Alkaline phosphatase; TG, Triglyceride; TP, total protein; TCHO, total cholesterol.” (Revised version: page 7, lines 148-149)

  1. Figures S1-S4 are a bit hard to read, please make sure to provide high-resolution images in the revised version of the manuscript.

Response: Thank you for pointing this out. We uploaded our original image in a different way and replaced it with a clearer image. (Supplementary materials (Supp Figs 1-4).PPT)

  1. While using "KEGG" for the first time, please include its full name.

Response: Thank you very much for your careful correction. We have expanded the full name of KEGG as follows: “Kyoto Encyclopedia of Genes and Genomes (KEGG)”. (Revised version: page 2, line 32)

  1. In the title of section 2.7, please replace "aggravated" with a more specific word, such as "increased".

Response: Thank you for your careful reading and professionalism in your suggestions. We have replaced “aggravated” with“increased.” (Revised version: page 13, line 232)

  1. In the discussion, while summarizing the major findings of this study, please also mention the corresponding figures/figure panels.

Response: Thank you for your professional suggestions. We have updated the “Discussion” section in the revision, and added more details relevant to the corresponding figures and tables as follows:

“The liver tissue of mice in the exposed group showed a large number of fatty vacuoles, inflammatory foci, and tumor nodules (Fig. 1A).” (Revised version: page 16, lines 268-269)

“From the above data (Table 1), we know that coexposure promotes disordered lipid metabolism in the liver, excessive accumulation of fat, steatohepatitis, and the deterioration of the microenvironment in the liver.” (Revised version: page 16, lines 284-286)

“Consistent with the literature report, the inflammatory score of the pathological staining patterns of mice in the coexposure group in this experiment was indeed the highest (Fig. 1B).” (Revised version: page 17, lines 289-291)

“We also identified the top ten differentially expressed genes when compared to the control group. At the end of the experiment, we observed significant differences in gene expression among the high-fat diet, nano-DEN diet, and coexposure diet groups when compared to the normal diet (Fig. 2). The coexposure group exhibited a larger number of liver cancer-related genes with more significant changes compared to the high-fat group and nano-DEN group (Fig.3, Supplementary Materials 1-4). This suggests that coexpo-sure may have the potential to promote or inhibit the expression of key transcription factors, thereby accel-erating the development of liver cancer.” (Revised version: page 17, lines 291-297)

“Compared with the nano-DEN group, the differential genes of the cancer region in the liver tissue of coex-posed mice were more enriched in the IL-17 signaling pathway, TGF-β signaling pathway, and PPAR sig-naling pathway (Fig.4).” (Revised version: page 17, lines 299-301)

“The above results suggest that coexposure exacerbated the high expression of genes associated with in-flammatory pathways in mice livers and promoted the malignant transformation of fatty liver disease into cancer. In our comprehensive data and visual analysis, we have identified distinct alterations in inflamma-tion-related pathways, hypoxia-related pathways, and aging-related pathways that exhibit pronounced sig-nificance in the progression from liver injury to hepatocellular carcinoma. Considering the pivotal protein components associated with inflammation factors (COX-2 and β-catenin), hypoxia (HIF-1α), and aging-related (p16), we regarded these as indicators for evaluating the extent of liver carcinogenesis. Therefore, we performed IHC staining for COX-2, β-catenin, HIF-1α, and p16 proteins to further substantiate our in-ferences from pathological aspects (Fig. 5).” (Revised version: page 17, lines 301-310)

We sincerely appreciate the reviewer's professional suggestions. In response, we have thoroughly reorganized our discussion section, adding substantial content to improve the logical flow and comprehensiveness of the information presented. These revisions aim to enhance the clarity and understanding of our study.

  1. In line 364, please include the full name of "SPF".

Response: Thank you for pointing this out. As reviewer’s suggestion, we have added the full name of “SPF” as followed: “Specific Pathogen Free (SPF)” (Revised version: page 19, line 358)

  1. Comments on the Quality of English Language.

Thank you for your professional suggestions. As reviewer’s suggestion, we have made some revisions to the wording in our manuscript. We have replaced "coexposuregroup" with "coexposure group" (Revised version: page 7, line 143)

  1. In lines 158-162, it is suggested to split and reorganize this long sentence into shorter ones.

Thank you for your professional suggestions. With your suggestion, we have rewritten the 2.3 section in the revision. Here are the sentences before and after modifications:

Before: “To maximize the screening of gene expression associated with these three diet conditions at the molecular level to investigate the effect of different dietary conditions on the alteration of normal liver, we considered high-fat diet, nanoDEN, and coexposure as separate conditions and attempted to search significant differences in gene expression from each of them, and derived possible patterns of differences by comparison with the control group. ” 

After: “The objective of this study was to maximize the screening of gene expression in liver tissues of mice under these three diet conditions. Then, we attempted to search significant differences in gene expression from each of them, and compared these differences with the control group to derive potential patterns. ” (Revised version: page 7, lines 151-153)

Your professional insights and suggestions are truly invaluable, whether it is ensuring grammatical accuracy or improving the clarity and comprehensibility of our manuscript.

We sincerely appreciate this invaluable opportunity you have given us!

Reviewer 2 Report

Comments and Suggestions for Authors

In this study, Li et al. investigate the impact of co-exposure to a high-fat diet and nano-diethylnitrosamine (nano-den)on the progression of fatty liver disease into liver cancer. They found that co-exposure to a high-fat diet and nano-DEN altered the tumor microenvironment and accelerated the progression of fatty liver disease into liver cancer. The liver transcriptomics analysis revealed that the expression levels of inflammatory, fatty, and fibrosis-related factors were elevated in the co-exposure group compared to the groups exposed to nano-DEN or a high-fat diet alone. The study also found that co-exposure aggravated the high expression of genes related to the carcinomatous pathway and accelerated the formation of the tumor microenvironment. The immunohistochemical staining results showed significant increases in abnormal protein changes related to inflammation, proliferation, aging, and hypoxia in mouse liver tissues. 

The results are intriguing; however, the following concerns need to be addressed:

  1. One of the major concerns of this study is novelty. It has been demonstrated before that this co-exposure promotes carcinogenesis and has a macrophage aspect to the pathology (PMID: PMID: 33304918). This study does not significantly improve upon this finding.
  2. There is a lack of mechanistic studies in this article. Although the authors have some indicators through their sequencing studies, there aren’t any experimental validations of mechanisms of co-exposure effects on hepatic cells.
  3. Many hepatocarcinoma cell lines of human origin could be tested for the effect on human cells for the translatability of their findings. Such in vitro studies are warranted.
  4. The authors need to explain better the statistical analysis and the number of samples in every experiment.
  5. The introduction is too long.
Comments on the Quality of English Language

There are several instances of grammatical errors and awkward phrasing throughout the manuscript that need to be addressed. I recommend professional proofreading from a native English speaker.

Author Response

Reply to Reviewer #2: 

We appreciated the reviewer’s effort of improving our manuscript. We have considered each of the suggestions in turn and have rewritten the manuscript accordingly to the reviewer’s comments.

With your suggestion, we have updated the “Result” section and “Discussion” section in the revision

  1. One of the major concerns of this study is novelty. It has been demonstrated before that this co-exposure promotes carcinogenesis and has a macrophage aspect to the pathology (PMID: 33304918). This study does not significantly improve upon this finding.

Response: Thank you for pointing out the deficiencies in our paper and recommending a relevant English article that aligns with our research. We have studied this article, and although our research topic is similar to this article, our focus on mechanism exploration is different. This article concludes based on techniques such as immunohistochemistry staining, immunofluorescence staining, and in vitro experiments, following the HE staining of relevant proteins. The results are very convincing.

At first, we had the same approach. However, when designing the experiments, we couldn't help but raise a question: what is the basis of selecting specific proteins? Is there an alternative approach that can comprehensively and rapidly screen for the most significant alterations influenced by the three dietary conditions, providing us with direction for subsequent research endeavors?

Therefore, unlike this article, in our study, we first utilized transcriptomics analysis to compare with the control group and identify significantly different transcription factors. We then screened closely related signaling pathways and analyzed the pathological changes at the molecular level. Furthermore, to validate our gene-level analysis results, we performed IHC staining on mouse liver tissues. We attempted to analyze the expression of relevant proteins from a pathological perspective to further support our hypotheses.

In our data analysis, we identified changes in inflammation-related pathways, hypoxia-related pathways, and aging-related pathways that were particularly evident in the progression of liver injury to hepatocellular carcinoma. Therefore, we selected these three interconnected pathways as the primary focus of our current study.

Based on this foundation and in conjunction with relevant literature, we chose specific proteins associated with inflammation (COX-2 and β-catenin), hypoxia (HIF-1α), and senescence (p16) in hepatocytes as indicators. We then conducted an analysis of the expression of these proteins to provide further support for our inferences drawn from a pathological perspective. Then, we performed IHC staining to investigate the expression of proteins (COX-2, β-catenin, HIF-1α, and p16) associated with the region of liver injury. As a result, we confirmed the theme of our manuscript on the ability of simultaneous administration of nano-DEN and high fat to accelerate liver tumorigenesis from both molecular and pathological aspects.

We believe that this method of rapidly retrieving relevant cancer gene pathways using biomedical engineering has a certain uniqueness. In the future of computer science advancement, perhaps this method can improve efficiency. We believe that this approach stands out as one of the key highlights of our research study.

Thank you for pointing out this question. In order to emphasize the novelty of this research, we have added more details of the reasonings in the “discussion” section as followed:

“We also identified the top ten differentially expressed genes when compared to the control group. At the end of the experiment, we observed significant differences in gene expression among the high-fat diet, nano-DEN diet, and coexposure diet groups when compared to the normal diet (Fig. 2). The coexposure group exhibited a larger number of liver cancer-related genes with more significant changes compared to the high-fat group and nano-DEN group (Fig.3, Supplementary Materials 1-4). This suggests that coexposure may have the potential to promote or inhibit the expression of key transcription factors, thereby accelerating the development of liver cancer.” (Revised version: page 17, lines 289-295)

“The above results suggest that coexposure exacerbated the high expression of genes associated with inflammatory pathways in mice livers and promoted the malignant transformation of fatty liver disease into cancer. In our comprehensive data and visual analysis, we have identified distinct alterations in inflammation-related pathways, hypoxia-related pathways, and aging-related pathways that exhibit pronounced significance in the progression from liver injury to hepatocellular carcinoma. Considering the pivotal protein components associated with inflammation factors (COX-2 and β-catenin), hypoxia (HIF-1α), and aging-related (p16), we regarded these as indicators for evaluating the extent of liver carcinogenesis. Therefore, we performed IHC staining for COX-2, β-catenin, HIF-1α, and p16 proteins to further substantiate our inferences from pathological aspects (Fig. 5).” (Revised version: page 17, lines 299-308)

In addition, we also added more relevant details about the rationale behind this analytical approach. This addition also serves as a better continuation of the fourth question (Q4) you raised.

In the “Materials and Methods” section, we have included additional details regarding the analytical method we employed, namely, whole-transcriptome analyses (RNA-seq). This technique involves a comprehensive examination of the entire transcriptome, allowing for a global view of gene expression patterns. The expanded description provides insights into the specific steps followed during the RNA-seq analysis, including library preparation, sequencing, read alignment, quality control, and differential expression analysis. These additions offer a more comprehensive understanding of the methodology used in our study. Thank you for your professionally probing questions.

  1. There is a lack of mechanistic studies in this article. Although the authors have some indicators through their sequencingstudies, there aren’t any experimental validations of mechanisms of co-exposure effects on hepatic cells. 

Response: Thank you for pointing out the limitations and shortcomings of our experiment. The mechanisms of co-exposure effects on hepatic cells involved in tumor development are quite complex and frequently cross-talk, requiring further time and effort to gradually prove our preliminary inferences step by step. There are many approaches that can be employed for mechanistic, such as in vitro experiments, knockdown or knockout studies, pathway analysis, and animal models. While we understand your concern about the lack of experimental validations on the mechanisms involved, we deliberately make more efforts in the last two approaches, sequencing analysis for this particular study and animal study because of the limited conditions. In this study, our main focus was on investigating the co-exposure effects of certain substances on hepatic cells using sequencing studies as indicators. As a result, we utilized bioinformatics tools to identify potential pathways and molecular networks associated with the co-exposure effects. Then we focus on analyzing gene expression data, protein-protein interaction databases, and pathway enrichment analysis, carefully selecting the most representative proteins based on a thorough literature review, and subsequently examining the mechanisms of protein function through extensive analysis of published research.

However, the suggestion you proposed is indeed warranted, and we appreciate your suggestions.  

In our upcoming research, we will gradually unravel the complex mechanisms behind the co-exposure effects on hepatic cells, ultimately providing a more comprehensive understanding of the underlying processes by combining more different approaches.

Once again, thank you for your professional advice.

  1. Many hepatocarcinoma cell lines of human origin could be tested for the effect on human cells for the translatability of their findings. Such in vitro studies are warranted.

Response: Thank you very much for your professional guidance and constructive feedback. Indeed, it is warranted to conduct experiments using hepatic cell lines or primary cells, or gene knockout for theoretical validation in vitro. In order to simulate the pathological changes caused by daily unhealthy eating habits and due to our experimental design focusing primarily on the field of biomedical engineering, we only utilized transcriptome analysis to identify representative differentially expressed genes that may play important roles in liver cancer formation. Currently, we only verified the relevant genes' specificity of the relevant genes through animal experiments. It provided preliminary evidence for our inferences at the pathological level. Although we did not perform a vitro study in this experiment, your suggestion is very necessary. The existing work showed that we have initially proved our point, and we would subsequently add these experimental data one by one to argue our inference in many aspects. In the following research, the study you suggested is indispensable to validate and reproduce the conclusions of our primitive study, making our subsequent experimental approach clearer and more refined.

Thanks again for your practical suggestions to complete our article!

  1. The authors need to explain better the statistical analysisand the number of samples in every experiment.

Response: Thank you for your professional suggestions. As the reviewer suggested, we have reworked the “Materials and Methods” section to make it easier for the reader to understand, and thank you again for your suggestions!

Here is the revised version:

“The experimental data are presented as the means ± SEM, and each group in the experiment consists of five samples (N=5). There are a total of four groups.” (Revised version: page 22, lines 425-426)

  • “4.5. Sample Collection and Quality Control

Simultaneously with the pathological examination group, another portion of each liver tissue sample was processed for whole-transcriptome analyses. Immediately after dissection, the liver tissue was cut into small chunks, approximately the size of a soybean. Each chunk was rapidly placed into pre-labeled vials liquid nitrogen to ensure immediate freezing for subsequent whole-transcriptome analyses. Total RNA was extracted from liver and tumor tissues collected from a total of 20 mice.  The quality of RNA samples was evaluated using the Agilent Bioanalyzer to ensure they met the experimental requirements.” (Revised version: page 20, lines 375-383)

  • “4.6. RNA Library Preparation and Sequencing Platform and Method

RNA libraries were prepared following the standard protocols provided by Illumina. This involved converting the RNA samples into libraries suitable for sequencing. Libraries were sequenced on Illumina HiS-eq 4000 platforms (Origingene, China). The reference genome used was sourced from the Ensembl database, with the genome version being GRCm38, and the gene annotation information was based on Ensemble 92. The high-quality sequencing reads obtained after quality control were processed using STAR software to align the read sequences with the specified reference genome (GRCm38) for the study species, which was the mouse (Mus musculus).” (Revised version: page 20, lines 384-391)

  • “4.7. Identification of Differentially Expressed Genes (DEGs)

DESeq2 was used for analyzing differential gene expression, enabling the identification of genes that exhibited statistically significant differences between the two sample groups. Differentially expressed genes (DEGs) were identified using DESeq2 (p value < 0.05 and | log2-fold-change | > 0.585) [46].” (Revised version: pages 20-21, lines 392-395)

  • “4.8. KEGG Pathway Enrichment Analysis

   The R package "clusterprofiler" was used to conduct KEGG pathway enrichment analysis, which helped identify the enrichment of differentially expressed genes in metabolic pathways and biological functions [47]. By utilizing the KEGG database, the genes from the samples were categorized based on their involvement in specific pathways or functional categories. The differential genes between pairwise groups, with one sample serving as the control, were then visualized on KEGG pathway maps, showcasing the an-notated pathways specific to these differential genes. The calculation was performed using Fisher's exact test. To control the false discovery rate (FDR) during the calculations, the Benjamini-Hochberg (BH) method was employed for multiple testing. The corrected p-values were set at 0.05, and KEGG pathways that met this criterion were defined as significantly enriched in differentially expressed genes.” (Revised version: page 21, lines 396-405)

  • “4.9. Data Deposition and Accessibility

   The data generated from this study has been deposited into the CNGB Sequence Archive, the database of the China National GeneBank [48]. It has been assigned the accession number CNP0003645, making it accessible to other researchers for validation and replication of the research findings [49].” (Revised version: page 21, lines 406-409)

  1. The introduction is too long.

Response: Thank you for your pertinent and objective criticism. As the reviewer suggested, we have rewritten the “Introduction” section as followed:

“The process of liver cancer is complex and its main risk factors include alcohol consumption, diabetes, obesity, and dietary exposures [2].” (Revised version: page 3, lines 44-45)

“Long-term consumption of a high-fat diet (HFD) has been shown to disrupt liver metabolism and promote lipid accumulation, resulting in the phosphorylation of insulin signaling transcription pathways, insulin resistance, and altered expression of Akt, GLUT4, and lipogenic genes [5, 6]. Insulin resistance triggers the breakdown of hepatic fat, with free fatty acids generated during this breakdown stimulating the expression of factors involved in cholesterol synthesis and fatty acid synthesis [7, 8].” (Revised version: page 3, lines 52-57)

“Excessive lipid accumulation in the liver leads to a hypoxic microenvironment within the tissue, activating hypoxia-inducible factor-1 (HIF-1) and promoting the expression of vascular endothelial growth factor to counteract hypoxia. Furthermore, it stimulates the aging of hepatic stellate cells and facilitates hepatic fibrosis [12].” (Revised version: page 3, lines 61-64)

“Encoded by the CTNNB1 gene, β-catenin forms part of the cadherin complex on the cell surface. Dysregulation of its activity is associated with hepatocellular carcinoma and other liver diseases. Notably, β-catenin participates in various signaling pathways, including TGF-β signaling, which promotes cell proliferation.” (Revised version: page 4, lines 72-75)

“These results indicate that β-catenin may activate multiple signaling pathways, resulting in the progression from fatty liver disease to liver cancer. In the previous study, we compared the mouse liver cancer model induced by nitrosamine and its nanopreparation and discovered that the nanopreparation rapidly induced liver cancer with significant efficacy and no side effects [25].” (Revised version: page 4, lines 83-86)

“Long-term excessive intake of high-fat food is a well-known unhealthy eating habit associated with liver fatigue, metabolic disorders, and an elevated risk of developing fatty liver. Similarly, the prolonged consumption of nitrite-rich pickled food represents another detrimental habit that inflicts irreversible damage on the liver, thereby facilitating a more rapid and facilitated onset of liver cancer. Unfortunately, the ex-tended duration required for lesion formation and the absence of symptoms affecting daily life make it easy to overlook the effects of pickled food consumption, inadvertently accelerating the occurrence of liver cancer without awareness.” (Revised version: page 4, lines 87-93)

The introduction, materials & methods, results, and figures of this manuscript have been thoroughly revised. With your patient guidance, we have incorporated numerous explanatory and analytical sections into the manuscript, as well as made revisions to the illustrations and references in the article. Your meticulous review, professional advice, clear insights, and point-by-point suggestions have greatly enhanced the rigor of our article. And the English of our manuscript has also been checked by an English lecturer Eric Newman working in New London Public School.

We sincerely thank you for your dedicated help!

Round 2

Reviewer 2 Report

Comments and Suggestions for Authors

The authors have provided a comprehensive textual rebuttal of the raised concerns. However, without the requested studies the gaps still remain (mechanisms and human cell line data) and are significant to be ignored.